# LOGICAL MESSAGE PASSING NETWORKS WITH ONE-HOP INFERENCE ON ATOMIC FORMULAS

**Zihao Wang & Yangqiu Song**
CSE, HKUST
Hong Kong SAR
{zwanggc,yqsong}@cse.ust.hk

**Ginny Y. Wong & Simon See**
NVIDIA AI Technology Center (NVATIC), NVIDIA
Santa Clara, USA
{gwong,ssee}@nvidia.com

## ABSTRACT

Complex Query Answering (CQA) over Knowledge Graphs (KGs) has attracted a lot of attention to potentially support many applications. Given that KGs are usually incomplete, neural models are proposed to answer the logical queries by parameterizing set operators with complex neural networks. However, such methods usually train neural set operators with a large number of entity and relation embeddings from the zero, where whether and how the embeddings or the neural set operators contribute to the performance remains not clear. In this paper, we propose a simple framework for complex query answering that decomposes the KG embeddings from neural set operators. We propose to represent the complex queries into the query graph. On top of the query graph, we propose the Logical Message Passing Neural Network (LMPNN) that connects the *local* one-hop inferences on atomic formulas to the *global* logical reasoning for complex query answering. We leverage existing effective KG embeddings to conduct one-hop inferences on atomic formulas, the results of which are regarded as the messages passed in LMPNN. The reasoning process over the overall logical formulas is turned into the forward pass of LMPNN that incrementally aggregates local information to finally predict the answers' embeddings. The complex logical inference across different types of queries will then be learned from training examples based on the LMPNN architecture. Theoretically, our query-graph representation is more general than the prevailing operator-tree formulation, so our approach applies to a broader range of complex KG queries. Empirically, our approach yields a new state-of-the-art neural CQA model. Our research bridges the gap between complex KG query answering tasks and the long-standing achievements of knowledge graph representation learning. Our implementation can be found at https://github.com/HKUST-KnowComp/LMPNN.

## 1 INTRODUCTION

Knowledge Graphs (KG) are essential sources of factual knowledge supporting downstream tasks such as question answering (Zhang et al., 2018; Sun et al., 2020; Ren et al., 2021). Answering logical queries is a complex but important task to utilize the given knowledge (Ren & Leskovec, 2020; Ren et al., 2021). Modern Knowledge Graphs (KG) (Bollacker et al., 2008; Suchanek et al., 2007; Carlson et al., 2010), though on a great scale, is usually considered incomplete. This issue is well known as the Open World Assumption (OWA) (Ji et al., 2021). Representation learning methods are employed to mitigate the incompleteness issue by learning representations from the observed KG triples and generalizing them to unseen triples (Bordes et al., 2013; Trouillon et al., 2016; Sun et al., 2018; Zhang et al., 2019; Chami et al., 2020). When considering logical queries over *incomplete* knowledge graphs, the query answering models are required to not only predict the unseen knowledge but also execute logical operators, such as conjunction, disjunction, and negation (Ren & Leskovec, 2020; Wang et al., 2021b).

Recently, neural models for Complex Query Answering (CQA) have been proposed to complete the unobserved knowledge graph and answer the complex query simultaneously. These models aim to address complex queries that belong to an important subset of the first-order queries. Formally speaking, the complex queries are Existentially quantified First Order queries and has a single free

variable (EFO-1) (Wang et al., 2021b) containing logical conjunction, disjunction, and negation (Ren & Leskovec, 2020). The EFO-1 queries are transformed in the forms of operator trees, e.g., relational set projection, set intersection, set union, and set complement (Wang et al., 2021b). The key idea of these approaches is to represent the entity set into specific embedding spaces (Ren & Leskovec, 2020; Zhang et al., 2021; Chen et al., 2022). Then, the set operators are parameterized by neural networks (Ren & Leskovec, 2020; Amayuelas et al., 2022; Bai et al., 2022). The strict execution of the set operations can be approximated by learning and conducting continuous mappings over the embedding spaces.

It is observed by experiments that classic KG representation (Trouillon et al., 2016) can easily outperform the neural CQA models in one-hop queries even though the neural CQA models model the one-hop projection with complex neural networks (Ren & Leskovec, 2020; Amayuelas et al., 2022; Bai et al., 2022). One possible reason is that the neural set projection is sub-optimal in modeling the inherent relational properties, such as symmetry, asymmetry, inversion, composition, etc, which are sufficiently discussed in KG completion tasks and addressed by KG representations (Trouillon et al., 2016; Sun et al., 2018). On the other hand, Continuous Query Decomposition (CQD) (Arakelyan et al., 2021) method searches for the best answers with a pretrained KG representation. The logical inference step is modeled as an optimization problem where the continuous truth value of an Existential Positive First Order (EPFO) query is maximized by altering the variable embeddings. However, the speed and the performance of inference heavily rely on the optimization algorithm. It also assumes that the embeddings of entities and relations can reflect higher-order logical relations, which is not generally assumed in existing knowledge graph representation models. Moreover, it is unclear whether CQD can achieve good performance on complex queries with negation operators [1].

In this paper, we aim to answer complex EFO-1 queries by equipping pretrained KG representations with logical inference power. First, we formulate the EFO-1 KG queries as Disjunctive Normal Form (DNF) formulas and propose to represent the conjunctive queries in the form of query graphs. In the query graph, each edge is an atomic formula that contains a predicate with a (possible) negation operator. For each one-hop atomic formula, we use the pretrained KG representation to infer the intermediate embeddings given the neighboring entity embedding, relation embedding, direction information, and negation information. We show that the inference can be analytically derived for the KG representation formulation. The results of one-hop atomic formula inference are interpreted as the logical messages passed from one node to another. Based on this mechanism, we propose a Logical Message Passing Neural Network (LMPNN), where node embeddings are updated by one Multi-Layer Perceptron (MLP) based on aggregated logical messages. LMPNN coordinates the local logical message by pretrained knowledge graph representations and predicts the answer embedding for a complex EFO-1 query. Instead of performing on-the-fly optimization over the query graph as CQD (Arakelyan et al., 2021), we parameterize the query answering process as the forward pass of LMPNN which is trained from the observed KG query samples.

Extensive experiments show that our approach is a new state-of-the-art neural CQA model, in which only one MLP network and two embedding vectors are trained. Interestingly, we show that the optimal number of layers of the LMPNN is the largest distance between the free variable node and the constant entity nodes. This makes it easy to generalize our approach to complex queries of arbitrary complexity. Hence, our approach bridges the gap between complex KG query answering tasks and the long-standing achievements of knowledge graph representation learning.

## 2 RELATED WORKS

**Knowledge graph representation.** Representing relational knowledge is one of the long-standing topics in representation learning. Knowledge graph representations aim to predict unseen relational triples by representing the discrete symbols in continuous spaces. Various algebraic structures (Bordes et al., 2013; Trouillon et al., 2016; Sun et al., 2018; Ebisu & Ichise, 2018; Zhang et al., 2019) are applied to represent the relational patterns (Sun et al., 2018) and different geometric spaces (Chami et al., 2020; Cao et al., 2022) are explored to capture the hierarchical structures in knowledge graphs. Therefore, entities and relations in large knowledge graphs can be efficiently represented in a continuous space.

---

[1]Existing empirical evaluations are all conducted on queries without negation (Arakelyan et al., 2021)

**Neural complex query answering.** Most existing works treat the complex queries as operator trees (Ren et al., 2020; Ren & Leskovec, 2020; Wang et al., 2021b). The query types that can be answered are extended from existential positive first-order (EPFO) queries (Ren et al., 2020; Choudhary et al., 2021; Arakelyan et al., 2021) to the existential first-order (Ren & Leskovec, 2020; Zhang et al., 2021; Bai et al., 2022), or more specifically EFO-1 queries (Wang et al., 2021b). In a neural CQA model, the entity sets are represented by various forms, including probabilistic distributions (Ren & Leskovec, 2020; Choudhary et al., 2021; Bai et al., 2022), geometric shapes (Ren et al., 2020; Zhang et al., 2021), and fuzzy-logic-inspired representations (Chen et al., 2022). In contrast to knowledge graph representations, the relation projections between sets are usually modeled by complex neural networks, including multi-layer perceptron (Ren & Leskovec, 2020), MLP-mixer (Amayuelas et al., 2022), or even transformers (Bai et al., 2022). However, their performances on one-hop queries are shown to be worse than the state-of-the-art but simple knowledge graph representation (Trouillon et al., 2016). Other works compiled the queries into the graphs and then solve queries with graph neural networks (Daza & Cochez, 2020; Liu et al., 2022). In contrast to this work, existing investigations only focused on EPFO queries and require to train the entire GNN from zero. Notably, knowledge graph representations also provide effective signals when answering complex queries. Specifically, CQD (Arakelyan et al., 2021) uses the KG representation to calculate the continuous truth value of an EPFO logical formula with the logical $t$-norms. Then, the embeddings are optimized to maximize the continuous truth value. The optimization can be applied in the embedding space as well as the symbolic space. Our experiments show that this method performs badly on complex queries with logical negation, see Section 4.3.

## 3 PRELIMINARIES

In this section, we formally introduce the knowledge graph and related model-theoretic concepts. These concepts are helpful when we define the DNF formulation of EFO-1 queries in Section 4. Then, we introduce the abstract interface of knowledge graph representation, which is useful in defining one-hop inference in Section 5.

**Model-theoretic concepts for knowledge graphs.** A first-order language $\mathcal{L}$ is specified by a triple $(\mathcal{F}, \mathcal{R}, \mathcal{C})$ where $\mathcal{F}$, $\mathcal{R}$, and $\mathcal{C}$ are sets of symbols for functions, relations, and constants, respectively. A knowledge graph is specified under the language $\mathcal{L}_{\mathcal{KG}}$, where function symbol set $\mathcal{F} = \emptyset$ and relation symbols in $\mathcal{R}$ denote binary relations. A knowledge graph $\mathcal{KG}$ is an $\mathcal{L}_{\mathcal{KG}}$-structure given the entity set $\mathcal{V}$, where each constant $c \in \mathcal{C} = \mathcal{V}$ is also an entity and each relation $r \in \mathcal{R}$ is a set $r \subseteq \mathcal{V} \times \mathcal{V}$. We say $r(t_1, t_2) = \text{True}$ when $(t_1, t_2) \in r$. A knowledge graph is usually defined by the relation triple set $\mathcal{E} = \{(h, r, t)\}$, where $h$ and $t$ are entities such that $(h, t) \in r$. The Open World Assumption (OWA) means only a subset of $\mathcal{E}$ can be observed. The observed knowledge graph is denoted by $\mathcal{KG}_{\text{obs}}$. A *term* is either a constant or a variable. And an *atomic formula* is either $r(t_1, t_2)$ or $\neg r(t_1, t_2)$ where $t_1$ and $t_2$ are terms and $r$ is a relation. In the following parts, we use $a.$ to denote an atomic formula. Then the first order formula can be inductively defined by adding *connectives* (conjunction $\wedge$, disjunction $\vee$, and negation $\neg$) to atomic formulas and *quantifiers* (existential $\exists$ and universal $\forall$) to variables. The formal definition of first-order formulas can be found in Marker (2006). A variable is bounded when associated with a quantifier, otherwise, it is free.

### 3.1 KNOWLEDGE GRAPH REPRESENTATIONS

Our approach relies upon the following abstract interface of knowledge graphs. Given the head entity embedding $\boldsymbol{h}$, relation embedding $\boldsymbol{r}$, and tail entity embedding $\boldsymbol{t}$, a knowledge graph representation is able to produce a continuous truth value $\psi(\boldsymbol{h}, \boldsymbol{r}, \boldsymbol{t})$ in $[0, 1]$ of the embedding triple $(\boldsymbol{h}, \boldsymbol{r}, \boldsymbol{t})$. In the symbolic space, whether $(h, t) \in r$ is indicated by the $\{0, 1\}$ truth value of $r(h, t)$. In the embedding space, $\psi(\boldsymbol{h}, \boldsymbol{r}, \boldsymbol{t})$ indicates the "probability" that $(h, t) \in r$. Hence, this definition is a continuous relaxation of the $\{0, 1\}$ truth value.

Each knowledge graph representation has a scoring function $\phi(h, r, t)$, which can be based on a similarity function or a distance function. It is easy to convert such functions into $\psi$ by applying the Sigmoid function with necessary scaling and shift. For example, the scoring function of ComplEx (Trouillon et al., 2016) embedding is.

$$\phi(\boldsymbol{h}, \boldsymbol{r}, \boldsymbol{t}) = \text{Re}(\langle \boldsymbol{h} \otimes \boldsymbol{r}, \bar{\boldsymbol{t}} \rangle), \tag{1}$$

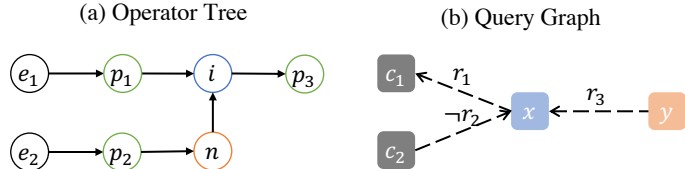

Figure 1: The operator tree representation (a) and query graph representation (b) of an examplar complex query in Ren & Leskovec (2020). The logical formula of this query is given by $r_1(x, c_1) \wedge \neg r_2(c_2, x) \wedge r_3(y, x)$. For shorthand, this query is denoted as INP. The symbols about relations and terms are consistent in the query graph representation. In the operator tree representation, $c_1$ and $c_2$ are represented by anchor node operators $e_1$ and $e_2$. Relation $r_1$ and $r_3$ are represented by projection node $p_1$ and $p_3$. Relation $r_2$ is jointly represented by projection node $p_2$ and negation node $p_n$. The fact that $x$ is connected to all other nodes is represented by intersection operator $i$.

where $\otimes$ denotes the element-wise complex number multiplication and $\langle x, y \rangle$ is the complex inner product. Re extracts the real part of a complex number. Then, the truth value of which can be computed by

$$\psi(\boldsymbol{h}, \boldsymbol{r}, \boldsymbol{t}) = \sigma(\phi(\boldsymbol{h}, \boldsymbol{r}, \boldsymbol{t})), \qquad (2)$$

where $\sigma$ is the sigmoid function. This truth value function is used in Arakelyan et al. (2021) with logic $t$-norms. In the context of knowledge graph representation learning, the entity embeddings $\boldsymbol{h}, \boldsymbol{t}$ are usually related to specific entity symbols in a look-up table. In this work, we assume the embedding vector is related to not only specific entities but also variables.

## 4 EFO-1 QUERIES AND QUERY GRAPHS

Without loss of generality, we consider the logical formulas under the disjunctive normal form. Then, we define the Existential First Order queries with a single free variable (EFO-1).

**Definition 1.** Given a knowledge graph $\mathcal{KG}$, an EFO-1 query $Q$ is formulated as the first-order formula in the following disjunctive normal form,

$$Q(y, x_1, ..., x_m) = \exists x_1, ..., \exists x_m \left[ a_{11} \wedge a_{12} \wedge \cdots \wedge a_{1n_1} \right] \vee \cdots \vee \left[ a_{p1} \wedge a_{p2} \wedge \cdots \wedge a_{pn_p} \right], \quad (3)$$

where $y$ is the only free variable and $x_i, 1 \leq i \leq m$ are $m$ existential variables. $a_{ij}, 1 \leq i \leq p, 1 \leq j \leq n_p$ are atomic formulas with constants and variables $y, x_1, \ldots, x_m$. $a_{ij}$ can be either negated or not.

To answer the EFO-1 queries, one is expected to identify the answer set $A[Q, \mathcal{KG}]$. $A[Q, \mathcal{KG}]$ is the set of entities such that $a \in A[q, \mathcal{KG}]$ if and only if $Q(y = a, x_1, ..., x_m) = $ True.

Moreover, since $Q$ is given in the disjunctive normal form, let us consider

$$Q(y, x_1, ..., x_m) = CQ_1(y, x_1, ..., x_m) \vee \cdots \vee CQ_p(y, x_1, ..., x_m), \qquad (4)$$

where $CQ_i = \exists x_1, ... \exists x_m a_{i1} \wedge a_{i2} \wedge \cdots \wedge a_{in_i}$ is a conjunctive query. It is easy to see that $A[Q, \mathcal{KG}] = \cup_{i=1}^{p} A[CQ_i, \mathcal{KG}]$. Therefore, solving $A[Q, \mathcal{KG}]$ is equivalent to solving the answer sets for all conjunctive queries.

### 4.1 QUERY GRAPH FOR CONJUNCTIVE QUERIES

For each conjunctive query, the constant entities and variables are closely related by the atomic formulas. To emphasize the dependencies between entities and variables, we propose to use the query graph where the terms are nodes connected by the atomic formulas. Each node in the query graph is either a constant symbol or a free or existential variable. Each edge in the query graph represents an atomic formula containing both relation and negation information.

Figure 1 shows our query graph representation and the operator tree representation (Wang et al., 2021b) for a typical query type defined in Ren & Leskovec (2020). We see that the query graph is more concise than the operator tree. First, we can see that the nodes and edges have different meanings in operator trees and query graphs. In the operator trees representation, each node is an operator denoting a set operation, whose output can be fed into other set operators. When using the complex query answering models with the operator tree, the information flows from leaf to root,

which is unidirectional. However, for the query graph, the messages are passed bi-directionally through each edge as we will show in Section 6. In Figure 1, the central node $x$ receives messages from all neighbor nodes.

## 4.2 EXPRESSIVENESS OF DEFINITION 1

Our definition is theoretically broader than all existing discussions. The definition in (Wang et al., 2021b), though widely adapted and discussed in the existing literature, has implicit assumptions because they are proposed to predict the answers by neural operators. It is assumed that (1) the Skolemization process can always convert the query into a tree of set operators, and (2) all leaves of the operator tree are entities rather than variables. A counterexample that can be expressed by Definition 1 but cannot be represented by operator trees is shown in Appendix A.

## 4.3 LIMITATION OF OPTIMIZATION-BASED METHODS FOR NEGATED QUERIES

Our definition accepts the atomic formulas with negation operation. Therefore, It can be seen as a natural extension of the definitions in CQD (Arakelyan et al., 2021). Moreover, we extended CQD to negation queries with the continuous truth value with fuzzy logical negator (see Appendix B). The extended method is named CQD(E), and the results are compared in Table 1. We could see that the performance of CQD(E) is much less effective on negation queries. We conjecture that the landscape of the objective function, i.e., the continuous truth values of the complex formula with negation, can be non-convex. So the optimization problem is inherently harder. The non-convexity objective function is discussed in Appendix B.1.

## 5 ONE-HOP INFERENCE ON ATOMIC FORMULAS

As shown in Figure 1, each edge in a query graph is an atomic formula containing the information of neighboring entities, relation, and logical negation, which are all crucial for predicting the answers. We propose to encode such entity, relation, and logical negation information by one-hop inference that maximizes the continuous truth value of the (negated) atomic formula. Let $\rho$ be the logical message encoding function of four input parameters, including neighboring entity embedding, relation embedding, direction information ($h2t$ or $t2h$), and logical negation information (0 for no negation and 1 for with negation). The goal of this section is to properly define $\rho$.

Moreover, inference on one-hop atomic formula is much easier compared to that on the entire complex EFO-1 query graph, as discussed in Section 4.3. We also provide the closed-form expression of $\rho$ for the knowledge graph embedding we used in this paper.

## 5.1 ONE-HOP INFERENCE IN NON-NEGATED ATOMIC FORMULAS

The first situation is to infer the head embedding $\hat{h}$ given the tail embedding $t$ and relation embedding $r$ on a non-negated atomic formula. We formulate the inference task in the form of *continuous truth value maximization*:

$$\hat{h} = \rho(t, r, t2h, 0) := \underset{x \in \mathcal{D}}{\arg\max}\, \psi(x, r, t), \tag{5}$$

where $\mathcal{D}$ is the search domain for the embedding. Similarly, the tail embedding $\hat{t}$ can be infered given head embedding $h$ and relation embedding $r$, that is,

$$\hat{t} = \rho(h, r, h2t, 0) := \underset{x \in \mathcal{D}}{\arg\max}\, \psi(h, r, x). \tag{6}$$

## 5.2 ONE-HOP INFERENCE IN NEGATED ATOMIC FORMULAS

To extend the definition for non-negated atomic formulas to negated formulas, one only need to compute the continuous truth value of a negated atomic formula by the fuzzy logic negator (Hájek,

(a) Logical Message Passing with One Hop Inference
(Query Graph View)

(b) Logical Message Passing Networks
(Node Embedding View)

Figure 2: An illustration of the two-stage procedures of logical message passing neural networks: (a) passing the logical messages across the graph; (b) updating the node embedding with the aggregated information with an MLP network.

2013), that is, $\psi(\boldsymbol{h}, \neg\boldsymbol{r}, \boldsymbol{t}) = 1 - \psi(\boldsymbol{h}, \boldsymbol{r}, \boldsymbol{t})$. Then the estimation of head and tail embeddings is related to the following inference problems

$$\hat{\boldsymbol{h}} = \rho(\boldsymbol{t}, \boldsymbol{r}, t2h, 1) := \arg\max_{\boldsymbol{x}\in\mathcal{D}} \psi(\boldsymbol{x}, \neg\boldsymbol{r}, \boldsymbol{t}) = \arg\max_{\boldsymbol{x}\in\mathcal{D}} \left[1 - \psi(\boldsymbol{x}, \boldsymbol{r}, \boldsymbol{t})\right], \tag{7}$$

$$\hat{\boldsymbol{t}} = \rho(\boldsymbol{h}, \boldsymbol{r}, h2t, 1) := \arg\max_{\boldsymbol{x}\in\mathcal{D}} \psi(\boldsymbol{h}, \neg\boldsymbol{r}, \boldsymbol{x}) = \arg\max_{\boldsymbol{x}\in\mathcal{D}} \left[1 - \psi(\boldsymbol{h}, \boldsymbol{r}, \boldsymbol{x})\right]. \tag{8}$$

This optimization-based approach is similar to CQD discussed in Section 4.3, but it is more reliable since atomic formulas are what we used to train the knowledge graph representation. Specifically, the objectives in Eq. (5) and Eq. (6) are eventually the likelihood of positive samples and those in Eq. (7) and Eq. (8) are the likelihood of negative samples. These objectives are widely used to learn the representations with negative sampling.

**Closed-form message encoding function $\rho$ with pretrained KG representation.** We have already defined $\rho$ with optimization problems. Moreover, the closed-form expression of $\rho$ can be (approximately) derived in many cases given two facts about the knowledge graph representations: (1) the scoring function of knowledge graph representation is usually as simple as the inner product or distance. More details about constructing closed-form $\rho$ for these two types of scoring functions are discussed in Appendix D.1; (2) the sigmoid function outside the scoring function $\phi$ makes the final truth value zero or one only if the output of the scoring function is sufficiently small or large. We identify the closed-form approximation of $\rho$ for ComplEx (Trouillon et al., 2016) and other five different KG representations in Appendix C and D, which allows fast computation logical messages used in Section 6.

## 6 LOGICAL MESSAGE PASSING NEURAL NETWORKS

In this section, we propose a Logical Message Passing Neural Network (LMPNN) to bridge the one-hop inference proposed in Section 5 and complex query answering defined in Section 4. As a variation of the message-passing neural network (Gilmer et al., 2017; Xu et al., 2018), LMPNN has two stages: (1) each node passes a message to all its neighbors; (2) each node aggregates the received messages and updates its latent embedding. Figure 2 illustrates how those two stages work. Then the final layer embedding for the free variable node can be used to predict the answer entities.

### 6.1 LOGICAL MESSAGE PASSING OVER THE QUERY GRAPH

We use the message encoding function $\rho$ to compute the messages passed from node to node. Figure 2 (a) demonstrates the logical message passing with blue arrows. Each node receives the message from all its neighboring nodes.

### 6.2 NODE EMBEDDINGS IN QUERY GRAPH AND UPDATING SCHEME

Let $n$ be a node and $\boldsymbol{z}_n^{(l)}$ be the embedding of $n$ at the $l$-th layer. We discuss how to compute the $\boldsymbol{z}_n^{(l)}$ from the input layer $l = 0$ to latent layers $l > 0$. When $l = 0$, $\boldsymbol{z}_n^{(0)}$ falls into one of three situations.

(1) For an entity node $e$, $\boldsymbol{z}_e^{(0)}$ is looked up from the pretrained knowledge graph representation. (2) For an existential variable node $x_i$, we assign an learnable embedding $\boldsymbol{z}_{x_i}^{(0)} = \boldsymbol{v}_x$. (3) For a free variable node $y$, we assign another learnable embedding $\boldsymbol{z}_y^{(0)} = \boldsymbol{v}_y$. We set that all existential variables $x_i$ share one $\boldsymbol{v}_x$, for simplicity.

At the $l$-th layer, $\boldsymbol{z}_n^{(l)}$ can be computed by updating the aggregated information from the $(l-1)$-th layer. Specifically, let $\mathcal{N}(n)$ be the neighbor set of node $n$ in the query graph. For each neighbor node $v \in \mathcal{N}(n)$, one can obtain its embedding $\boldsymbol{z}_v^{(l-1)} \in \mathcal{D}$, the relation $r_{v \to n} \in \mathcal{R}$, the direction $\mathrm{D}_{v \to n} \in \{h2t, t2h\}$, and the negation indicator $\mathrm{Neg}_{v \to n} \in \{0, 1\}$. Then, the embedding $\boldsymbol{z}_n^{(l)}, l \geq 1$ is computed by an MLP network after the summation of the aggregated information, that is,

$$\boldsymbol{z}_n^{(l)} = \mathrm{MLP}^{(l)} \left[ \epsilon \boldsymbol{z}_n^{(l-1)} + \sum_{v \in \mathcal{N}(n)} \rho \left( \boldsymbol{z}_v^{(l-1)}, r_{v \to n}, \mathrm{D}_{v \to n}, \mathrm{Neg}_{v \to n} \right) \right], \qquad (9)$$

where $\epsilon$ is a hyperparameter. To feed the complex vector of ComplEx (Trouillon et al., 2016) into the MLP network, the real and imaginary vectors of one complex embedding are concatenated and regarded as one feature vector. The formulation in Eq. (9) is similar to the Graph Isomorphic Networks (Xu et al., 2018) except that the logical messages passed are encoded by $\rho$ from the pretrained KG representation. Trainable $\boldsymbol{v}_x$ and $\boldsymbol{v}_y$ are unrelated to any specific entity.

## 6.3 LEARNING LMPNN FOR COMPLEX QUERY ANSWERING

To train the neural network, we apply the Noisy Contrastive Estimation (NCE) loss for ranking tasks proposed in (Ma & Collins, 2018). Let $\{(a_i, q_i)\}_{i=1}^n$ be the positive data samples, where $a_i \in A[q_i, \mathcal{KG}]$. Our optimization involves $K$ uniformly sampled noisy answers from the entity set. The NCE objective is:

$$L_{NCE} = \frac{1}{n} \sum_{i=1}^n \log \left[ \frac{\exp \left[ \cos(\boldsymbol{a}_i, \boldsymbol{z}(q_i))/T \right]}{\exp \left[ \cos(\boldsymbol{a}_i, \boldsymbol{z}(q_i))/T \right] + \sum_{k=1}^K \exp \left[ \cos(\boldsymbol{z}_k, \boldsymbol{z}(q_i))/T \right]} \right], \qquad (10)$$

where $\boldsymbol{a}_i$ is the embedding of positive answer $a_i$ and $\boldsymbol{z}_k$ is the embedding of the noisy entity samples. $z(q_i)$ indicates the embedding of the free variable in $q_i$ at the final layer of LMPNN. $T$ is a hyperparameter. This objective is optimized by stochastic gradient descent.

## 6.4 ANSWERING COMPLEX QUERIES WITH LMPNN

We discuss two ways to retrieve answers for general DNF queries in Definition 1: (a) A two-step approach as the previous works (Ren et al., 2020; Ren & Leskovec, 2020), where the free variable embedding for each sub conjunctive query are estimated, the answer entities are then ranked by the minimal distance (or maximal similarity) against free variable embeddings from multiple sub conjunctive queries. (2) We transform all disjunctions in the formula to conjunctions, then one query graph is sufficient for solving the transformed query. The answer set of a transformed query is a strict subset of the originanl answer. For simplicity, we use the second way to solve disjunctive queries in this paper, though it may lead to sub-optimal performance. Then, we discuss how to answer conjunctive queries with LMPNN.

**Conjunctive query graph of arbitrary size.** We apply LMPNN to the query graph of a given conjunctive query $Q$. A sufficient condition to produce a correct answer is that the free variable node has received messages from all the entity nodes after the forward passing through LMPNN layers. Let the largest distance between entity nodes and the free variable node be $L$. Then, we apply the LMPNN layers $L$ times to ensure all messages from entity nodes are successfully received by the free variable node. The prediction of answer embedding $\boldsymbol{z}(Q)$ is given by the free variable embedding at the final layer, i.e., $\boldsymbol{z}(Q) = \boldsymbol{z}_y^{(L)}$. We propose to use the cosine similarity between $\boldsymbol{z}(Q)$ and the pretrained entity embeddings to rank the entities and then retrieve answers.

Since $L$ is not determined, we assume all $L$ layers share the **same** MLP layer. Hence, the only trainable parameter in LMPNN is one MLP network and two embeddings for existential and free variables. Our experiments on different query types show that the single MLP network has strong generalizability to LMPNN of different depths.

Table 1: MRR results of different CQA models on three KGs. $A_\mathrm{P}$ represents the average score of EPFO queries and $A_\mathrm{N}$ represents the average score of queries with negation. The boldface indicates the best results for each KG.

| KG | Model | 1P | 2P | 3P | 2I | 3I | PI | IP | 2U | UP | 2IN | 3IN | INP | PIN | PNI | $A_\mathrm{P}$ | $A_\mathrm{N}$ |
|---|---|---|---|---|---|---|---|---|---|---|---|---|---|---|---|---|---|
| FB15K | BetaE | 65.1 | 25.7 | 24.7 | 55.8 | 66.5 | 43.9 | 28.1 | 40.1 | 25.2 | 14.3 | 14.7 | 11.5 | 6.5 | 12.4 | 41.6 | 11.8 |
| | ConE | 73.3 | 33.8 | **29.2** | 64.4 | 73.7 | **50.9** | 35.7 | **55.7** | **31.4** | 17.9 | 18.7 | 12.5 | 9.8 | 15.1 | 49.8 | 14.8 |
| | Q2P | 82.6 | 30.8 | 25.5 | 65.1 | 74.7 | 49.5 | 34.9 | 32.1 | 26.2 | 21.9 | 20.8 | 12.5 | 8.9 | 17.1 | 46.8 | 16.4 |
| | *(Using pretrained KG representation)* | | | | | | | | | | | | | | | | |
| | CQD(E) | **89.4** | 27.6 | 15.1 | 63.0 | 65.5 | 46.0 | 35.2 | 42.9 | 23.2 | 0.2 | 0.2 | 4.0 | 0.1 | **18.4** | 45.3 | 4.6 |
| | LMPNN | 85.0 | **39.3** | 28.6 | **68.2** | **76.5** | 46.7 | **43.0** | 36.7 | **31.4** | **29.1** | **29.4** | **14.9** | **10.2** | 16.4 | **50.6** | **20.0** |
| FB15K -237 | BetaE | 39.0 | 10.9 | 10.0 | 28.8 | 42.5 | 22.4 | 12.6 | 12.4 | 9.7 | 5.1 | 7.9 | 7.4 | 3.6 | 3.4 | 20.9 | 5.4 |
| | ConE | 41.8 | 12.8 | **11.0** | 32.6 | 47.3 | **25.5** | 14.0 | **14.5** | **10.8** | 5.4 | 8.6 | **7.8** | 4.0 | 3.6 | 23.4 | 5.9 |
| | Q2P | 39.1 | 11.4 | 10.1 | 32.3 | 47.7 | 24.0 | 14.3 | 8.7 | 9.1 | 4.4 | 9.7 | 7.5 | **4.6** | 3.8 | 21.9 | 6.0 |
| | *(Using pretrained KG representation)* | | | | | | | | | | | | | | | | |
| | CQD(E) | **46.7** | 10.3 | 6.5 | 23.1 | 29.8 | 22.1 | 16.3 | 14.2 | 8.9 | 0.2 | 0.2 | 2.1 | 0.1 | **6.1** | 19.8 | 1.7 |
| | LMPNN | 45.9 | **13.1** | 10.3 | **34.8** | **48.9** | 22.7 | **17.6** | 13.5 | 10.3 | **8.7** | **12.9** | 7.7 | **4.6** | 5.2 | **24.1** | **7.8** |
| NELL | BetaE | 53.0 | 13.0 | 11.4 | 37.6 | 47.5 | 24.1 | 14.3 | 12.2 | 8.5 | 5.1 | 7.8 | 10.0 | 3.1 | 3.5 | 24.6 | 5.9 |
| | ConE | 53.1 | 16.1 | 13.9 | 40.0 | **50.8** | 26.3 | 17.5 | 15.3 | 11.3 | 5.7 | 8.1 | 10.8 | 3.5 | 3.9 | 27.2 | 6.4 |
| | Q2P | 56.5 | 15.2 | 12.5 | 35.8 | 48.7 | 22.6 | 16.1 | 11.1 | 10.4 | 5.1 | 7.4 | 10.2 | 3.3 | 3.4 | 25.5 | 6.0 |
| | *(Using pretrained KG representation)* | | | | | | | | | | | | | | | | |
| | CQD(E) | **60.8** | 18.3 | 13.2 | 36.5 | 43.0 | **30.0** | 22.5 | **17.6** | 13.7 | 0.1 | 0.1 | 4.0 | 0.0 | **5.2** | 28.4 | 1.9 |
| | LMPNN | 60.6 | **22.1** | **17.5** | **40.1** | 50.3 | 28.4 | **24.9** | 17.2 | **15.7** | **8.5** | **10.8** | **12.2** | **3.9** | 4.8 | **30.7** | **8.0** |

# 7 EXPERIMENTS

In this section, we compare LMPNN with existing neural CQA methods and justify the important features of LMPNN with ablation studies. Our results show that LMPNN is a very strong method for answering complex queries.

## 7.1 EXPERIMENTAL SETTINGS

**Baselines.** We consider the neural complex query answering models for EFO-1 queries in recent three years, including BetaE (Ren & Leskovec, 2020), ConE (Zhang et al., 2021), and Q2P (Bai et al., 2022). The baseline results are obtained by training models with the code released by the authors under the suggested hyperparameters. Neural-symbolic ensemble models are implemented with the negation. Moreover, we also implement and report CQD (Arakelyan et al., 2021) with the same pretrained knowledge graph representation. We also compare more neural CQA models in Appendix E.

**Datasets.** We consider the widely used training and evaluation dataset in (Ren & Leskovec, 2020). It allows us to compare our results with existing methods directly. We compare the results on FB15k (Bordes et al., 2013), FB15k-237 (Toutanova et al., 2015), and NELL (Carlson et al., 2010).

**Evaluations.** The evaluation metric follows the previous works (Ren & Leskovec, 2020). For each query instance, we first rank all entities except those observed as easy answers based on their cosine similarity with the free variable embedding estimated by LMPNN. The rankings of hard answers are used to compute MRR for the given query instance. Then, we average the metrics from all query instances. In this paper, MRR is reported and compared.

**LMPNN Setting.** We use the ComplEx (Trouillon et al., 2016) checkpoints released by Arakelyan et al. (2021) in LMPNN to make a fair comparison to CQD (Arakelyan et al., 2021). More results about LMPNN with other six kinds of pretrained KG representations are also presented in the Appendix D. The rank of ComplEx is 1,000, and the epoch for the checkpoint is 100. For LMPNN, we use AdamW to train the MLP network. The learning rate is 1e-4, and the weight decay is 1e-4. The batch size is 1,024, and the negative sample size is 128, selected from $\{32, 128, 512\}$. The MLP network has one hidden layer whose dimension is 8,192 for NELL and FB15k, and 4,096 for FB15k-237. $T$ in the training objective is chosen as $0.05$ for FB15k-237 and FB15k and $0.1$ for NELL. $\epsilon$ in Eq (9) is chosen to be $0.1$. Reported results are averaged from 3 random experimental trials. All experiments of LMPNN are conducted on a single V100 GPU (16GB).

## 7.2 MAJOR RESULTS

Table 1 presents the MRR results of LMPNN and neural CQA baselines on answering EFO-1 queries over three KGs. It is found that LMPNN reaches the best performance on average for both EPFO and

Table 2: MRR results of different hyperparameter settings compared to the best combination.

| Model | 1P | 2P | 3P | 2I | 3I | PI | IP | 2U | UP | 2IN | 3IN | INP | PIN | PNI | $A_\mathrm{P}$ | $A_\mathrm{N}$ |
|---|---|---|---|---|---|---|---|---|---|---|---|---|---|---|---|---|
| KGE CAT | 30.5 | 6.8 | 6.9 | 7.8 | 7.8 | 6.2 | 6.4 | 8.4 | 6.3 | 3.0 | 2.8 | 5.6 | 2.3 | 1.7 | 9.7 | 3.1 |
| $\epsilon = 0$ | 45.5 | 12.6 | 9.7 | 33.6 | 47.1 | 11.1 | 17.1 | 14.0 | 10.0 | 8.8 | 11.6 | 7.3 | 3.5 | 2.7 | 22.3 | 6.8 |
| $\epsilon = 0.5$ | 43.9 | 11.9 | 9.7 | 30.5 | 42.3 | 18.2 | 14.8 | 13.8 | 9.6 | 7.3 | 10.3 | 7.0 | 3.8 | 5.4 | 21.6 | 6.7 |
| $L - 1$ | 45.5 | 6.8 | 7.8 | 34.1 | 47.5 | 11.6 | 6.3 | 13.9 | 6.1 | 8.6 | 11.7 | 5.6 | 2.9 | 3.9 | 20.0 | 6.5 |
| $L + 1$ | 45.3 | 12.6 | 10.1 | 33.1 | 46.4 | 20.4 | 16.3 | 13.7 | 10.0 | 8.0 | 10.9 | 7.5 | 4.3 | 5.2 | 23.1 | 7.2 |
| $L + 2$ | 45.6 | 13.0 | 10.1 | 32.9 | 45.8 | 21.3 | 17.7 | 13.4 | 10.0 | 7.9 | 10.9 | 7.7 | 4.2 | 5.2 | 23.3 | 7.2 |
| $L + 3$ | 45.4 | 13.0 | 10.1 | 32.4 | 45.2 | 22.1 | 17.4 | 13.3 | 10.0 | 7.7 | 10.9 | 7.7 | 4.2 | 5.2 | 23.2 | 7.1 |
| $T = 0.01$ | 43.9 | 12.1 | 9.7 | 31.9 | 46.5 | 18.3 | 15.2 | 13.5 | 10.2 | 6.9 | 11.2 | 7.2 | 4.6 | 4.3 | 22.4 | 6.8 |
| $T = 0.1$ | 45.6 | 12.7 | 9.9 | 33.7 | 47.1 | 22.0 | 17.1 | 14.0 | 10.0 | 8.8 | 11.5 | 7.3 | 4.4 | 5.4 | 23.6 | 7.5 |
| BEST CHOICE | 45.9 | 13.1 | 10.3 | 34.8 | 48.9 | 22.7 | 17.6 | 13.5 | 10.3 | 8.7 | 12.9 | 7.7 | 4.6 | 5.2 | 24.1 | 7.8 |

negation queries. Our results on negation queries indicate that the embedding estimation formulation with negated atomic formula proposed in Section 6.1 produces meaningful features.

Interestingly, LMPNN performs much better than the CQD on both EPFO and negation queries with the same pretrained knowledge graph representation. Our results show that the LMPNN is stronger than CQD in more complex queries, especially those with logical negation. Notably, our approach does not require any optimization in the inference time as in Arakelyan et al. (2021). It confirms again that LMPNN successfully leverages the representation power of knowledge graph representation simply by training an MLP.

## 7.3 ABLATION STUDY

In the ablation study, we conduct extensive experiments to justify the effects of four key factors of LMPNN, including (1) the logical message passing by one-hop inference; (2) the hyperparameter $\epsilon$ at each LMPNN layer; (3) the depth of LMPNN; (4) the hyperparameter $T$ at the noisy contrastive learning loss. All experiments of the ablation study are conducted on queries at FB15K-237.

To justify the effect of one-hop inference, we compare a baseline with logical messages computed by a linear transformation of the concatenation of the entity embedding, the relation embedding, a binary indicator for $h2t$ and $t2h$, and a binary indicator for negation. For example, for ComplEx embedding in 1,000-dimensional complex vector space, there are 2,000 parameters for entity embedding and 2,000 for relation embedding. The concatenation produces a feature of 4,002 dimensions. Then we use a linear transformation to transform this feature to 2000 dimensions so that the logical message can be used in Eq. (9). This baseline is denoted as KGE CAT. To justify the effect of the depth of LMPNN, we alter the depth of LMPNN based on its original depth $L$ into $L - 1$, $L + 1$, $L + 2$, and $L + 3$. The value $L$ is computed from the maximal distances between the free variable node and constant entity nodes in the query graph. Even in the $L - 1$ case, we keep the depth of LMPNN at least 1 to ensure the logical message is passed between nodes.

Table 2 shows the results of the ablation study, where the setting reported in Table 1 is indicated by BEST CHOICE. We note that BEST CHOICE uses one-hop inference on atomic formulas, $L$ LMPNN layers, $\epsilon = 0.1$, and $T = 0.05$ for FB15k-237. We find that KGE CAT performs poorly even though it contains the pretrained KG information, which indicates that one-hop inference is essential to answer complex queries. Meanwhile, $L - 1$ performs worse than BEST CHOICE since the information is not fully passed to the free variable node. And the worse performances of $L + 1$, $L + 2$, and $L + 3$ cases indicate that our definition for $L$ is reasonable. Moreover, $\epsilon$ and $T$ are also important to the best performance. Overall, one-hop inference on atomic formula is the most critical factor in the learning and inference process of LMPNN.

## 8 CONCLUSION

In this paper, we present LMPNN to answer complex queries, especially EFO-1 queries, over knowledge graphs. LMPNN achieves a strong performance by training one MLP network to aggregate the logical messages passed over the query graph. In the ablation study, we identify that the one-hop inference on atomic formulas based on a pretrained knowledge graph is critical to answering complex queries. Our research effectively bridges the gap between EFO-1 query answering tasks and the long-standing achievements of knowledge graph representation. In future work, our method can be combined with stronger knowledge graph representation techniques, as well as with neural-symbolic ensembles.

## 9  ACKONWLEDGEMENT

The authors of this paper were supported by the NSFC Fund (U20B2053) from the NSFC of China, the RIF (R6020-19 and R6021-20) and the GRF (16211520 and 16205322) from RGC of Hong Kong, the MHKJFS (MHP/001/19) from ITC of Hong Kong and the National Key R&D Program of China (2019YFE0198200) with special thanks to HKMAAC and CUSBLT, and the Jiangsu Province Science and Technology Collaboration Fund (BZ2021065). We also thank the support from NVIDIA AI Technology Center (NVAITC) and the UGC Research Matching Grants (RMGS20EG01-D, RMGS20CR11, RMGS20CR12, RMGS20EG19, RMGS20EG21, RMGS23CR05, RMGS23EG08).

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

## A  A COUNTEREXAMPLE FOR THE EXPRESIVENESS OF OPERATOR TREE REPRESENTATION

**Example 1.**  Given a citation network with authors, papers, and conferences, one query wants to find *ICLR authors with at least one collaborator*. It can be expressed in the format in Definition 1 as

$$q(a_1, a_2, p_1, p_2) = \exists a_2 \exists p_1 \exists p_2 \text{IsAuthor}(a_1, p_1) \wedge \text{InConf}(p_1, \text{ICLR})$$
$$\wedge \text{IsAuthor}(a_1, p_2) \wedge \text{IsAuthor}(a_2, p_2) \wedge \neg(a_1 = a_2).$$

We see that if we take $a_1$ as the answer node. $a_2$ and $p_2$ are leaves but not anchor entities. In this way, this query cannot be represented by the operator tree anchor nodes. However, this query can be represented in the query graph, see Figure 3.

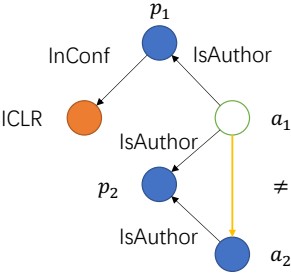

Figure 3: The query graph for the query in Example 1.

Then we discuss how to answer this query with LMPNN. It is easy to see that LMPNN can be applied to the query graph in Figure 3 once the $\neq$ is considered as the combination of predicate $eq$ (equality) and negation $\neg$.

To include $eq$, we only need to define the logical messages $\rho(\boldsymbol{a}_1, eq, 1)$ and $\rho(\boldsymbol{a}_2, eq, 1)$. According to the Proposition 2 in Appendix D.1, these two problems boil down to defining $\rho(\boldsymbol{a}, eq, 0) = f(\boldsymbol{a}, eq)$. By the semantics of "equality", equal terms shares the equal embedding. Therefore, the entity embedding which is equal to a given embedding $\boldsymbol{a}$ is just $f(\boldsymbol{a}, eq) = \boldsymbol{a}$. Then, $\rho(\boldsymbol{a}, eq, 0) = f(\boldsymbol{a}, eq) = \boldsymbol{a}$ and $\rho(\boldsymbol{a}_1, eq, 1) = -\boldsymbol{a}_1$, $\rho(\boldsymbol{a}_2, eq, 1) = -\boldsymbol{a}_2$. In this way, LMPNN is the first actionable approach to address the queries in Example 1.

## B  A NATURAL EXTENSION OF COMPLEX QUERY DECOMPOSITION (CQD) TO ANSWER NEGATION QUERIES

In this paper, we compare the optimization-based approach CQD (Arakelyan et al., 2021) by extending existing CQD with fuzzy logic negator. The extended version is denoted as CQD(E). For example, consider the logical formula INP query in the Figure 1, we could estimate the continuous truth value of the given logical formula $r_1(x, c_1) \wedge \neg r_2(c_2, x) \wedge r_3(y, x)$ as follows

$$TV_{\text{CQD(E)}}(\boldsymbol{x}, \boldsymbol{y}|\text{INP}) = \psi_{r_1}(\boldsymbol{x}, \boldsymbol{c}_1)\top [1 - \psi_{r_2}(\boldsymbol{c}_2, \boldsymbol{x})] \top \psi_{r_3}(\boldsymbol{y}, \boldsymbol{x}), \qquad (11)$$

where $\phi.$ are the continuous value of relations $r_1$, $r_2$, and $r_3$ and $\top$ is a $t$-norm. Then CQD(E) maximizes the continuous truth value $TV_{\text{CQD(E)}}(\boldsymbol{x}, \boldsymbol{y}|\text{INP})$ to obtain the "best" variable embeddings $\boldsymbol{x}$ and $\boldsymbol{y}$ as Arakelyan et al. (2021).

### B.1  NON-CONVEX LANDSCAPE OF NEGATED COMPLEX QUERIES

In this part, we show that the negator in fuzzy logic introduces non-convexity. Let $x$ be an optimizable variable in the 1D interval $I$ and $\phi_1(x)$ and $\phi_2(x)$ be two continuous truth value of two atomic

formula $a_1$ and $a_2$, respectively. They are convex functions over $I$. Consider the conjunctive query $a_1 \wedge \neg a_2$. The continuous truth value is

$$J(x) = \phi_1(x)\top[1 - \phi_2(x)]. \tag{12}$$

Consider an example with convex $\phi_1(x)$ and $\phi_2(x)$. Let $\top$ is the product $t$-norm, and $\phi_1(x) = 1-x^2$ and $\phi_2(x) = 1 - (x - 0.3)^2$ for $x \in [0, 1]$. Then $J(x)$ turns to be non-convex as shown in Figure 4.

## C  CLOSED-FORM LOGICAL MESSAGE BY COMPLEX

In this section, we derive the closed-form logical message encoding function for ComplEx embedding (Trouillon et al., 2016). The scoring function of ComplEx is

$$\phi(\boldsymbol{h}, \boldsymbol{r}, \boldsymbol{t}) = \mathrm{Re}(\langle \boldsymbol{h} \otimes \boldsymbol{r}, \bar{\boldsymbol{t}} \rangle). \tag{13}$$

We expand the complex embeddings to real vectors $\boldsymbol{h} = \boldsymbol{h}_r + i\boldsymbol{h}_i$, $\boldsymbol{r} = \boldsymbol{r}_r + i\boldsymbol{r}_i$, $\boldsymbol{t} = \boldsymbol{t}_r + i\boldsymbol{t}_i$. Then the scoring function is

$$\phi(\boldsymbol{h}, \boldsymbol{r}, \boldsymbol{t}) = \mathrm{Re}(\langle \boldsymbol{h} \otimes \boldsymbol{r}, \bar{\boldsymbol{t}} \rangle) \tag{14}$$

$$= \langle \boldsymbol{r}_r \otimes \boldsymbol{h}_r - \boldsymbol{r}_i \otimes \boldsymbol{h}_i, \boldsymbol{t}_r \rangle + \langle \boldsymbol{r}_r \otimes \boldsymbol{h}_i + \boldsymbol{r}_i \otimes \boldsymbol{h}_r, \boldsymbol{t}_i \rangle \tag{15}$$

$$= \langle \boldsymbol{r}_r \otimes \boldsymbol{t}_r + \boldsymbol{r}_i \otimes \boldsymbol{t}_i, \boldsymbol{h}_r \rangle + \langle \boldsymbol{r}_r \otimes \boldsymbol{t}_i - \boldsymbol{r}_i \otimes \boldsymbol{t}_r, \boldsymbol{h}_i \rangle. \tag{16}$$

Since $-\boldsymbol{r}_i = \bar{\boldsymbol{r}}_i$ under the complex conjugate, then,

$$\phi(\boldsymbol{h}, \boldsymbol{r}, \boldsymbol{t}) = \langle \boldsymbol{r}_r \otimes \boldsymbol{t}_r - \bar{\boldsymbol{r}}_i \otimes \boldsymbol{t}_i, \boldsymbol{h}_r \rangle + \langle \boldsymbol{r}_r \otimes \boldsymbol{t}_i + \bar{\boldsymbol{r}}_i \otimes \boldsymbol{t}_r, \boldsymbol{h}_i \rangle \tag{17}$$

$$= \mathrm{Re}(\langle \boldsymbol{t} \otimes \bar{\boldsymbol{r}}, \bar{\boldsymbol{h}} \rangle). \tag{18}$$

Then, we optimize the continuous truth value of ComplEx given in Eq. (2) to derive the closed-form estimation of Eq. (5). We note that the embedding used in ComplEx is not strictly restricted in a domain set $\mathcal{D}$. Instead, the N3 regularization (Lacroix et al., 2018) is applied to the embedding as a *soft* constraint. Therefore, in our derivation of the close form solution, we also employ N3 regularization rather than hard constraint. Our first result is the following proposition.

**Proposition 1.** For ComplEx embedding, the logical message encoding function has the following closed form with respect to the complex embedding $\boldsymbol{r}$ and $\boldsymbol{t}$,

$$\rho(\boldsymbol{t}, \boldsymbol{r}, t2h, 0) = \frac{\bar{\boldsymbol{r}} \otimes \boldsymbol{t}}{\sqrt{3\lambda \|\boldsymbol{r} \otimes \bar{\boldsymbol{t}}\|}}. \tag{19}$$

*Proof.* We expand the optimization problem as follows,

$$\rho(\boldsymbol{t}, \boldsymbol{r}, t2h, 0) = \underset{\boldsymbol{x} \in \mathbb{C}^d}{\arg\max} \left\{ \mathrm{Re}(\langle \bar{\boldsymbol{r}} \otimes \boldsymbol{t}, \bar{\boldsymbol{x}} \rangle) - \lambda \|\boldsymbol{x}\|^3 \right\} \tag{20}$$

$$= \underset{\boldsymbol{x} \in \mathbb{C}^d}{\arg\max} \left\{ \langle \boldsymbol{r}_r \otimes \boldsymbol{t}_r - \bar{\boldsymbol{r}}_i \otimes \boldsymbol{t}_i, \boldsymbol{x}_r \rangle + \langle \boldsymbol{r}_r \otimes \boldsymbol{t}_i + \bar{\boldsymbol{r}}_i \otimes \boldsymbol{t}_r, \boldsymbol{x}_i \rangle - \lambda \left( \sqrt{\langle \boldsymbol{x}_r, \boldsymbol{x}_r \rangle + \langle \boldsymbol{x}_i, \boldsymbol{x}_i \rangle} \right)^3 \right\}. \tag{21}$$

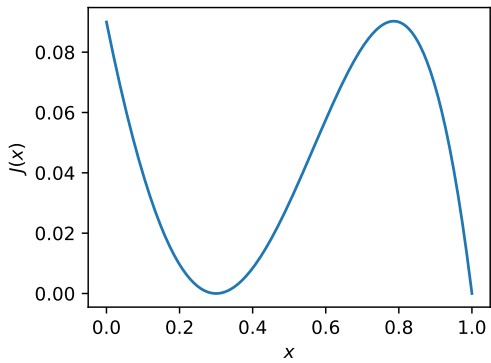

Figure 4: The landscape of continuous truth value becomes non-convex after negation.

Notice that $\boldsymbol{r}_r \otimes \boldsymbol{t}_r - \bar{\boldsymbol{r}}_i \otimes \boldsymbol{t}_i$ and $\boldsymbol{r}_r \otimes \boldsymbol{t}_i + \bar{\boldsymbol{r}}_i \otimes \boldsymbol{t}_r$ are the real and imaginary part of $\bar{\boldsymbol{r}} \otimes \boldsymbol{t}$. Let $\boldsymbol{s} = [\boldsymbol{r}_r \otimes \boldsymbol{t}_r - \bar{\boldsymbol{r}}_i \otimes \boldsymbol{t}_i, \boldsymbol{r}_r \otimes \boldsymbol{t}_i + \bar{\boldsymbol{r}}_i \otimes \boldsymbol{t}_r]$ be the real vector concatenated by the real and imageinary part of $\bar{\boldsymbol{r}} \otimes \boldsymbol{t}$. Also, let $\tilde{\boldsymbol{x}} = [\boldsymbol{x}_r, \boldsymbol{x}_i]$ be the real vector concatenated by the real and imageinary part of $\boldsymbol{x}$. Then the Eq. (21) is equivalent to the following optimization problem in the real space:

$$\max_{\tilde{\boldsymbol{x}} \in \mathbb{R}^{2d}} \underbrace{\langle \boldsymbol{s}, \tilde{\boldsymbol{x}} \rangle - \lambda \|\tilde{\boldsymbol{x}}\|_2^3}_{:= J}. \tag{22}$$

We note that $J$ is convex function over $\tilde{\boldsymbol{x}}$. To optimize $\tilde{\boldsymbol{x}}$, we optimize the unit direction $\boldsymbol{v}$ and length $\eta$ of $\tilde{\boldsymbol{x}}$, with rewriting $\tilde{\boldsymbol{x}} = \eta \boldsymbol{v}$. When $\eta$ is fixed the second term is also fixed, it is easy to see that the $\boldsymbol{v}^* = \boldsymbol{s} / \|\boldsymbol{s}\|_2$ maximizes the first term. Then we find the optimal $\eta$ by minizing the following objective for $\eta > 0$:

$$J = \|\boldsymbol{s}\|_2 \eta - \lambda \eta^3. \tag{23}$$

By letting $\frac{dJ}{d\eta} = \|\boldsymbol{s}\|_2 - 3\lambda\eta^2 = 0$, we derive the optimal $\eta^* = \sqrt{\frac{\|\boldsymbol{s}\|_2}{3\lambda}}$. Then, we have

$$\tilde{\boldsymbol{x}}^* = \eta^* \boldsymbol{v}^* = \frac{\boldsymbol{s}}{\sqrt{3\lambda\|\boldsymbol{s}\|_2}}. \tag{24}$$

Then, we identify the optimal real and imaginary part of $\boldsymbol{x}^*$ from $\tilde{\boldsymbol{x}}^*$, and thus recover the optimal $\boldsymbol{x}^*$. $\qquad\square$

Similarly, we derive the optimal closed-form expression of $\rho$ in all other cases:

$$\rho(\boldsymbol{h}, \boldsymbol{r}, h2t, 0) = \frac{\boldsymbol{r} \otimes \boldsymbol{h}}{\sqrt{3\lambda\|\boldsymbol{r} \otimes \boldsymbol{h}\|}}, \tag{25}$$

$$\rho(\boldsymbol{t}, \boldsymbol{r}, t2h, 1) = \frac{-\bar{\boldsymbol{r}} \otimes \boldsymbol{t}}{\sqrt{3\lambda\|\bar{\boldsymbol{r}} \otimes \boldsymbol{t}\|}}, \tag{26}$$

$$\rho(\boldsymbol{h}, \boldsymbol{r}, h2t, 1) = \frac{-\boldsymbol{r} \otimes \boldsymbol{h}}{\sqrt{3\lambda\|\boldsymbol{r} \otimes \boldsymbol{h}\|}}. \tag{27}$$

We note that the value of $\lambda$ is not determined. On the one hand, it can be of course a hyperparameter to discuss. In LMPNN application, we just let $3\lambda\| \cdot \| = 1$ and then all denominators in the closed-form expression are 1.

## D  CLOSED-FORM LOGICAL MESSAGES FOR KG REPRESENTATIONS

We demonstrate two general ways to construct closed-form logical messages function $\rho$ for LMPNN in the Appendix D.1. Then, we show six examples to illustrate how our approach constructs the closed form $\rho$ for various KG representations in the Appendix D.2.

Specifically, our constructions apply to two types of KG representations characterized by their scoring functions. The first type of KG representations uses *inner-product-based scoring functions* while the second type of KG representations uses *distance-based scoring functions*. Moreover, we provide six examples of KG representations, including RESCAL (Nickel et al., 2011), TransE (Bordes et al., 2013), DistMult (Yang et al., 2014), ComplEx (Trouillon et al., 2016), ConvE (Dettmers et al., 2018), and RotatE (Sun et al., 2018).

### D.1  TWO CONSTRUCTIONS

As discussed in Section 5, the closed-form logical message encoding function $\rho$ is the result of the closed-form solution of four one-hop inference problems (estimating the head or tail entity embedding with or without logical negation, see Eq. (5-8). This leads to four construction tasks. The major result of Appendix D.1 is Proposition 2. It shows that, with our constructions of two types of scoring

functions, closed-form solutions for four one-hop inference problems are actually dependent. Once one of four one-hop inference problems are approximately solved in the closed form, the other three one-hop inference problems are also solved approximately in the closed form.

**Simplification with reciprocal relations.** We simplify the four construction tasks into two tasks by introducing reciprocal relations. For each relation $r \in \mathcal{R}$, the reciprocal relation is $r^{-1} \in \mathcal{R}^{-1}$ but in the reversed direction. By introducing reciprocal relations $r^{-1}$ and training their embeddings $\boldsymbol{r}^{-1}$, the one-hop inference in the tail-to-head direction can be rewritten in the head-to-tail direction. Specifically, we have

$$\rho(\boldsymbol{t}, \boldsymbol{r}, t2h, 0) = \rho(\boldsymbol{t}, \boldsymbol{r}^{-1}, h2t, 0), \tag{28}$$

$$\rho(\boldsymbol{t}, \boldsymbol{r}, t2h, 1) = \rho(\boldsymbol{t}, \boldsymbol{r}^{-1}, h2t, 1). \tag{29}$$

Introducing reciprocal relations is shown to improve the performances of the link prediction tasks (Ruffinelli et al., 2020). We assume that the reciprocal relation embedding can be obtained, irrespective of being separately trained or analytically derived from the original relation embedding, such as ComplEx discussed in Appendix C. Then, it suffices to construct the closed-form solution for $\rho(\boldsymbol{h}, \boldsymbol{r}, h2t, 0)$ and $\rho(\boldsymbol{h}, \boldsymbol{r}, h2t, 1)$, and the rest two types of logical messages are naturally defined with reciprocal relation embeddings.

Then we construct the closed-form $\rho(\boldsymbol{h}, \boldsymbol{r}, h2t, 0)$ and $\rho(\boldsymbol{h}, \boldsymbol{r}, h2t, 1)$ for two types of KG embeddings, characterized by their scoring functions. We emphasize that the derivations below are only approximate estimations to keep the closed-form expression as simple as possible. However, empirical results show that these simple and approximate closed-form solutions can already be used in LMPNN.

**Type 1: inner-product-based scoring function.** The inner-product-based scoring function for a triple of embeddings $(\boldsymbol{h}, \boldsymbol{r}, \boldsymbol{t})$ is $\langle f(\boldsymbol{h}, \boldsymbol{r}), \boldsymbol{t} \rangle$, where $f$ is a binary function of the entity and relation embeddings and $\langle \cdot, \cdot \rangle$ is the inner product. The inner-product $\langle \cdot, \cdot \rangle$ can be defined in real or complex vector spaces. This scoring function is used in RESCAL (Nickel et al., 2011) DistMult (Yang et al., 2014), ComplEx (Trouillon et al., 2016), ConvE (Dettmers et al., 2018), etc.

When optimizing the embeddings, $l_2^q$ regularizations ($q = 2, 3$) are usually applied (Ruffinelli et al., 2020). Then we consider the following optimization problem:

$$\rho(\boldsymbol{h}, \boldsymbol{r}, h2t, 0) = \arg\max_{\boldsymbol{x}} \underbrace{\sigma\left(\langle f(\boldsymbol{h}, \boldsymbol{r}), \boldsymbol{x}\rangle\right) - \lambda\|\boldsymbol{x}\|_2^p}_{:=J_1}, \tag{30}$$

where hyperparameter $\lambda > 0$ is a regularization coefficient, $\sigma$ is the sigmoid function.

We note that $J_1$ is just the Lagrangian of the following maximization problem, and the $\lambda$ is the Langrangian multiplier

$$\max_{\|\boldsymbol{x}\|_2^q < \delta} \sigma\left(\langle f(\boldsymbol{h}, \boldsymbol{r}), \boldsymbol{x}\rangle\right), \tag{31}$$

where $\boldsymbol{x}$ is restricted inside a $\delta^{\frac{1}{q}}$-ball. Then we could conclude that

$$\arg\max_{\|\boldsymbol{x}\|_2^q < \delta} \sigma\left(\langle f(\boldsymbol{h}, \boldsymbol{r}), \boldsymbol{x}\rangle\right) = \arg\max_{\|\boldsymbol{x}\|_2^q < \delta} \langle f(\boldsymbol{h}, \boldsymbol{r}), \boldsymbol{x}\rangle = \delta^{\frac{1}{q}} \frac{f(\boldsymbol{h}, \boldsymbol{r})}{\|f(\boldsymbol{h}, \boldsymbol{r})\|_2}. \tag{32}$$

By altering the hyperparameter $\delta = \|f(\boldsymbol{h}, \boldsymbol{r})\|_2^p$, we could derive a simple result

$$\arg\max_{\boldsymbol{x}} \sigma\left(\langle f(\boldsymbol{h}, \boldsymbol{r}), \boldsymbol{x}\rangle\right) - \lambda\|\boldsymbol{x}\|_2^p \approx f(\boldsymbol{h}, \boldsymbol{r}). \tag{33}$$

Therefore, we define

$$\rho(\boldsymbol{h}, \boldsymbol{r}, h2t, 1) := f(\boldsymbol{h}, \boldsymbol{r}). \tag{34}$$

Similarly, for the $\rho(\boldsymbol{h}, \boldsymbol{r}, h2t, 1)$, we have

$$\rho(\boldsymbol{h}, \boldsymbol{r}, h2t, 1) = \arg\max_{\boldsymbol{x}} \left[1 - \sigma\left(\langle f(\boldsymbol{h}, \boldsymbol{r}), \boldsymbol{x}\rangle\right)\right] - \lambda\|\boldsymbol{x}\|_2^p \tag{35}$$

$$= \arg\max_{\boldsymbol{x}} \sigma\left(\langle -f(\boldsymbol{h}, \boldsymbol{r}), \boldsymbol{x}\rangle\right) - \lambda\|\boldsymbol{x}\|_2^p \tag{36}$$

$$\approx \arg\max_{\|\boldsymbol{x}\|_2^p < \delta} \langle -f(\boldsymbol{h}, \boldsymbol{r}), \boldsymbol{x}\rangle. \tag{37}$$

We conclude the closed-form solution as

$$\rho(\boldsymbol{h}, \boldsymbol{r}, h2t, 0) := -f(\boldsymbol{h}, \boldsymbol{r}). \qquad (38)$$

**Type 2: distance-based scoring function.** Another type of scoring functions for a triple of embeddings $(\boldsymbol{h}, \boldsymbol{r}, \boldsymbol{t})$ is $\gamma - \|f(\boldsymbol{h}, \boldsymbol{r}) - \boldsymbol{t}\|$, where $f$ follows the definition above and $\gamma$ is a margin. This scoring function is used in TransE (Bordes et al., 2013), RotatE (Sun et al., 2018), etc. Similarly, the $\|\boldsymbol{x}\|_2^q$ regularizations can also be considered (Ruffinelli et al., 2020).

$\rho(\boldsymbol{h}, \boldsymbol{r}, h2t, 0)$ can be computed by

$$\rho(\boldsymbol{h}, \boldsymbol{r}, h2t, 0) = \arg\max_{\boldsymbol{x}} \sigma\left(\gamma - \|f(\boldsymbol{h}, \boldsymbol{r}) - \boldsymbol{x}\|\right) - \lambda\|\boldsymbol{x}\|_2^p. \qquad (39)$$

With similar tricks, we transform the "soft" regularization into the "hard" constraint.

$$\arg\max_{\boldsymbol{x}} \sigma\left(\gamma - \|f(\boldsymbol{h}, \boldsymbol{r}) - \boldsymbol{x}\|\right) - \lambda\|\boldsymbol{x}\|_2^q \approx \arg\max_{\|\boldsymbol{x}\|_2^q < \delta}\left[\gamma - \|f(\boldsymbol{h}, \boldsymbol{r}) - \boldsymbol{x}\|\right], \qquad (40)$$

where $\delta$ is another hyperparameter. We set $\delta > \|f(\boldsymbol{h}, \boldsymbol{r})\|_2^q$, then the optimal solution is $f(\boldsymbol{h}, \boldsymbol{r})$, which summarizes

$$\rho(\boldsymbol{h}, \boldsymbol{r}, h2t, 0) := f(\boldsymbol{h}, \boldsymbol{r}). \qquad (41)$$

For the negated head-to-tail direction, the one-hop inference problem is

$$\arg\max_{\|\boldsymbol{x}\|_2^q < \delta}\left[1 - \sigma\left(\gamma - \|f(\boldsymbol{h}, \boldsymbol{r}) - \boldsymbol{x}\|\right)\right] \qquad (42)$$

$$= \arg\max_{\|\boldsymbol{x}\|_2^q < \delta} \|f(\boldsymbol{h}, \boldsymbol{r}) - \boldsymbol{x}\| = -\delta^{\frac{1}{q}} f(h, r). \qquad (43)$$

For simplicity, we choose

$$\rho(\boldsymbol{h}, \boldsymbol{r}, h2t, 1) := -f(\boldsymbol{h}, \boldsymbol{r}). \qquad (44)$$

Our constructions for two types of KG representations share a unified closed-form logical message once the function $f(\boldsymbol{h}, \boldsymbol{r})$ is given. In the following part $f$ is named as "forward" estimation function since it estimate the tail embeddings based on head and relation embedding in a forward direction. Therefore, we summarize the four types of logical messages used in LMPNN in the following proposition:

**Proposition 2.** For a KG representation of either Type 1 and Type 2, we could define four closed-form logical encoding functions with (1) relation embedding $\boldsymbol{r}$ and the corresponding reciprocal relation embedding $\boldsymbol{r}^{-1}$ and (2) the forward estimation function $f$ as follows:

$$\rho(\boldsymbol{h}, \boldsymbol{r}, h2t, 0) = f(\boldsymbol{h}, \boldsymbol{r}), \qquad (45)$$

$$\rho(\boldsymbol{h}, \boldsymbol{r}, h2t, 1) = -f(\boldsymbol{h}, \boldsymbol{r}), \qquad (46)$$

$$\rho(\boldsymbol{t}, \boldsymbol{r}, t2h, 0) = f(\boldsymbol{t}, \boldsymbol{r}^{-1}), \qquad (47)$$

$$\rho(\boldsymbol{t}, \boldsymbol{r}, t2h, 1) = -f(\boldsymbol{t}, \boldsymbol{r}^{-1}). \qquad (48)$$

## D.2  SIX KG REPRESENTATION EXAMPLES

Now it is ready to apply the Proposition 2 to six KG representations. For each KG representation, it is important to state its scoring function for triple $(h, r, t)$ and the relation parameterization. We assume the reciprocal relation embeddings are already trained.

Table 3 summarizes the information for RESCAL (Nickel et al., 2011) TransE (Bordes et al., 2013), DistMult (Yang et al., 2014), ComplEx (Trouillon et al., 2016), ConvE (Dettmers et al., 2018), and RotatE (Sun et al., 2018). We list the relation parameter $\boldsymbol{r}$, the essential one-hop inference function $\rho(\boldsymbol{h}, \boldsymbol{r}, h2t, 0) = f(\boldsymbol{h}, \boldsymbol{r})$, and the scoring function for each KG representation.

The scoring function of ComplEx (Trouillon et al., 2016) is not the exact inner-product in the complex vector space, but it can be reduced to the inner product in the real vector space and has already been discussed in Appendix C. We see that the Proposition 2 and Table 3 covers the results in Appendix C by letting the reciprocal embedding of $\boldsymbol{r}^{-1}$ be the complex conjugate $\bar{\boldsymbol{r}}$ of the original embedding $\boldsymbol{r}$.

Table 3: Closed-form forward estimation function $f$ for six KG representations. Closed-form logical message encoding function $\rho$ can be easily constructed with the closed-form $f$.

| KG Embedding | $\boldsymbol{r}$ parameters | $f(\boldsymbol{h}, \boldsymbol{r})$ | Scoring function |
|---|---|---|---|
| RESCAL (Nickel et al., 2011) | $W_r$ | $W_r \boldsymbol{h}$ | $\langle f(\boldsymbol{h}, \boldsymbol{r}), \boldsymbol{t} \rangle$ |
| TransE (Bordes et al., 2013) | $\boldsymbol{r}$ | $\boldsymbol{r} + \boldsymbol{h}$ | $\gamma - \|f(\boldsymbol{h}, \boldsymbol{r}) - \boldsymbol{t}\|$ |
| DistMult (Yang et al., 2014) | $\boldsymbol{r}$ | $\boldsymbol{r} \otimes \boldsymbol{h}$ | $\langle f(\boldsymbol{h}, \boldsymbol{r}), \boldsymbol{t} \rangle$ |
| ComplEx (Trouillon et al., 2016) | $\boldsymbol{r}$ | $\boldsymbol{r} \otimes \boldsymbol{h}$ | $\mathrm{Re}\langle f(\boldsymbol{h}, \boldsymbol{r}), \bar{\boldsymbol{t}} \rangle$ |
| ConvE (Dettmers et al., 2018) | $\omega, W$ | $\texttt{ReLU}(\texttt{vec}(\texttt{ReLU}([e_h; e_r] * \omega))W)$ | $\langle f(\boldsymbol{h}, \boldsymbol{r}), \boldsymbol{t} \rangle$ |
| RotatE (Sun et al., 2018) | $\cos\theta + i\sin\theta$ | $(\cos\theta + i\sin\theta) \otimes \boldsymbol{h}$ | $\gamma - \|f(\boldsymbol{h}, \boldsymbol{r}) - \boldsymbol{t}\|$ |

Table 4: Properties of six backbone KG representations on FB15-237. The data is released by Ruffinelli et al. (2020) on `https://github.com/uma-pi1/kge`. The dimension of each KG representation is listed in the bracket. We note that the dimensions for the complex vector embeddings indicates the trainable parameters. For example, each ComplEx embedding of 256D is a complex vector in $\mathbb{C}^{128}$ with 256 trainable parameters.

| KG Repr. | MRR | Hits@1 | Hits@3 | Hits@10 | Config file | Checkpoint |
|---|---|---|---|---|---|---|
| RESCAL (128D) | 0.356 | 0.263 | 0.393 | 0.541 | [download link] | [download link] |
| TransE (128D) | 0.313 | 0.221 | 0.347 | 0.497 | [download link] | [download link] |
| DistMult (256D) | 0.343 | 0.250 | 0.378 | 0.531 | [download link] | [download link] |
| ComplEx (256D) | 0.348 | 0.253 | 0.384 | 0.536 | [download link] | [download link] |
| ConvE (256D) | 0.339 | 0.248 | 0.369 | 0.521 | [download link] | [download link] |
| RotatE (256D) | 0.333 | 0.240 | 0.368 | 0.522 | [download link] | [download link] |

### D.3 PERFORMANCES OF LMPNN WITH DIFFERENT BACKBONE KG REPRESENTATIONS

The performances of LMPNN with six backbone KG representations are presented in Table 5. The LMPNN is trained in the suggested setting in the Section 7.3. The pretrain checkpoints of six backbone KG representations are obtained from Ruffinelli et al. (2020). The information of the performances of each KG representation is listed in Table 4.

It could be found that, LMPNN achieves descent performances with simple KG backbones of relatively low dimensions (128D and 256D). ConvE (256D) (Dettmers et al., 2018), DistMult (256D) (Yang et al., 2014), and ComplEx (256D) Trouillon et al. (2016) outperform BetaE (800D) on both EPFO and negation queries. All KG representation except TransE (128D) (Bordes et al., 2013) could outperform BetaE (800D) (Ren & Leskovec, 2020) on negation queries. We note that adjust the hyperparameters, i.e., embedding dimensions, to obtain more powerful KG representations could improve the results. However, this is beyond the scope of this paper.

## E NEURAL CQA BENCHMARK

In this section, we show that LMPNN (with ComplEx 2000D pretrained by (Arakelyan et al., 2021)) is the new state-of-the-art method among all neural CQA models. We include the following neural CQA baselines that can address the EFO-1 queries. Other models that cannot answer EFO-1 queries

Table 5: Comparison of LMPNN with different pretrained backbone KG representations on FB15k-237 queries.

| Model | 1P | 2P | 3P | 2I | 3I | PI | IP | 2U | UP | 2IN | 3IN | INP | PIN | PNI | $A_{\mathrm{P}}$ | $A_{\mathrm{N}}$ |
|---|---|---|---|---|---|---|---|---|---|---|---|---|---|---|---|---|
| TransE (128D) | 39.9 | 9.2 | 8.4 | 23.5 | 36.2 | 8.2 | 10.0 | 10.1 | 5.4 | 3.4 | 6.9 | 5.8 | 3.0 | 2.2 | 16.8 | 4.2 |
| RESCAL (128D) | 43.6 | 11.9 | 9.9 | 33.7 | 48.0 | 9.8 | 16.2 | 12.1 | 9.9 | 4.2 | 10.3 | 7.0 | 3.5 | 2.5 | 21.7 | 5.5 |
| ConvE (256D) | 42.5 | 12.3 | 10.5 | 30.6 | 43.8 | 9.6 | 13.0 | 11.8 | 7.6 | 5.2 | 9.8 | 7.1 | 3.8 | 3.5 | 20.2 | 5.9 |
| RotatE (256D) | 43.8 | 11.2 | 8.9 | 30.4 | 44.5 | 11.0 | 15.0 | 13.0 | 8.6 | 7.0 | 10.5 | 6.3 | 3.7 | 3.2 | 20.7 | 6.2 |
| DistMult (256D) | 43.6 | 11.2 | 9.5 | 32.2 | 46.3 | 18.1 | 15.1 | 13.0 | 9.3 | 6.1 | 10.5 | 6.6 | 4.1 | 4.2 | 22.0 | 6.3 |
| ComplEx (256D) | 44.4 | 11.7 | 9.3 | 32.4 | 46.4 | 18.1 | 15.7 | 13.0 | 9.4 | 6.0 | 10.7 | 6.8 | 4.1 | 4.0 | 22.3 | 6.4 |

Table 6: Benchmark comparison with neural CQA models on FB15k-237 queries.

| Model | 1P | 2P | 3P | 2I | 3I | PI | IP | 2U | UP | 2IN | 3IN | INP | PIN | PNI | $A_{\text{P}}$ | $A_{\text{N}}$ |
|-------|-----|------|------|------|------|------|------|------|------|-----|------|-----|-----|-----|------|-----|
| BETAE | 39.0 | 10.9 | 10.0 | 28.8 | 42.5 | 22.4 | 12.6 | 12.4 | 9.7 | 5.1 | 7.9 | 7.4 | 3.6 | 3.4 | 20.9 | 5.5 |
| ConE | 41.8 | 12.8 | 11.0 | 32.6 | 47.3 | 25.5 | 14.0 | 14.5 | 10.8 | 5.4 | 8.6 | 7.8 | 4.0 | 3.6 | 23.4 | 5.9 |
| MLP-Mix | 43.4 | 12.6 | 10.4 | 33.6 | 47.0 | 14.9 | 25.7 | 14.2 | 10.2 | 6.6 | 10.7 | 8.1 | 4.7 | 4.4 | 23.6 | 6.9 |
| Q2P | 39.1 | 11.4 | 10.1 | 32.3 | 47.7 | 24.0 | 14.3 | 8.7 | 9.1 | 4.4 | 9.7 | 7.5 | 4.6 | 3.8 | 21.9 | 6.0 |
| CQD | 46.7 | 10.3 | 6.5 | 23.1 | 29.8 | 22.1 | 16.3 | 14.2 | 8.9 | 0.2 | 0.2 | 2.1 | 0.1 | 6.1 | 19.8 | 1.7 |
| LMPNN | 45.9 | 13.1 | 10.3 | 34.8 | 48.9 | 22.7 | 17.6 | 13.5 | 10.3 | 8.7 | 12.9 | 7.7 | 4.6 | 5.2 | **24.1** | **7.8** |

Table 7: Benchmark comparison with neural CQA models on FB15k queries.

| Model | 1P | 2P | 3P | 2I | 3I | PI | IP | 2U | UP | 2IN | 3IN | INP | PIN | PNI | $A_{\text{P}}$ | $A_{\text{N}}$ |
|-------|-----|------|------|------|------|------|------|------|------|------|------|------|------|------|------|------|
| BETAE | 65.1 | 25.7 | 24.7 | 55.8 | 66.5 | 43.9 | 28.1 | 40.1 | 25.2 | 14.3 | 14.7 | 11.5 | 6.5 | 12.4 | 41.7 | 11.9 |
| ConE | 73.3 | 33.8 | 29.2 | 64.4 | 73.7 | 50.9 | 35.7 | 55.7 | 31.4 | 17.9 | 18.7 | 12.5 | 9.8 | 15.1 | 49.8 | 14.8 |
| MLP-Mix | 71.9 | 32.1 | 27.1 | 59.9 | 70.5 | 33.7 | 48.4 | 40.4 | 28.4 | 17.2 | 17.8 | 13.5 | 9.1 | 15.2 | 45.8 | 14.6 |
| Q2P | 82.6 | 30.8 | 25.5 | 65.1 | 74.7 | 49.5 | 34.9 | 32.1 | 26.2 | 21.9 | 20.8 | 12.5 | 8.9 | 17.1 | 46.8 | 16.2 |
| CQD | 89.4 | 27.6 | 15.1 | 63.0 | 65.5 | 46.0 | 35.2 | 42.9 | 23.2 | 0.2 | 0.2 | 4.0 | 0.1 | 18.4 | 45.3 | 4.6 |
| LMPNN | 85.0 | 39.3 | 28.6 | 68.2 | 76.5 | 46.7 | 43.0 | 36.7 | 31.4 | 29.1 | 29.4 | 14.9 | 10.2 | 16.4 | **50.6** | **20.0** |

are not compared (Ren et al., 2020; Choudhary et al., 2021; Liu et al., 2022). We tried to reproduce the results reported, and we note that different model applies to different knowledge graphs.

**BetaE (Ren & Leskovec, 2020):** Results are reproduced for FB15k-237, FB15k, and NELL.
**ConE (Zhang et al., 2021):** Results are reproduced for FB15k-237, FB15k, and NELL.
**MLP-Mix (Amayuelas et al., 2022):** Results are reproduced for FB15k-237, FB15k, and NELL.
**Q2P (Bai et al., 2022):** Results are reproduced for FB15k-237, FB15k, and NELL.
**FuzzQE (Chen et al., 2022):** Results on FB15k are missing. Results on FB15k-237 are not reproducible with the given code and suggested hyperparameters. Results on NELL are partially reproduced, so we report the results in the paper and reproduced by us.
**CQD (Arakelyan et al., 2021):** Results are reproduced on FB15k-237, FB15k, and NELL.

The results of FB15k-237, FB15k, and NELL are shown in Table 6, Table 7, and Table 8, respectively. We can see that LMPNN achieves the best performance among all neural complex query answering models.

## F    COMPARE TO SYMBOLIC INTEGRATION METHODS

Contextualized and symbolic information are shown to be effective to improve the neural models for both knowledge graph representation and complex query answering. For knowledge graph representation, neighboring information (Schlichtkrull et al., 2018; Wang et al., 2019; 2021a; Zhu et al., 2021) aggregated by graph neural networks of KG, external information (Xie et al., 2016a;b) by annotations, or even information from language models (Petroni et al., 2019; Liu et al., 2020) are also leveraged to make the knowledge graph representation more informative and effective. For complex query answering, neural models are enhanced with symbolic resoning (Zhu et al., 2022; Xu et al., 2022) that heavily search over the original symbolic space (Zhu et al., 2022) or its approximations (Cohen et al., 2020; Xu et al., 2022). Unlike neural CQA models whose operations are always in the embedding space of fixed size, the size of the intermediate states for symbolic reasoning grows with the number of the entity sets, such as the fuzzy sets used in (Zhu et al., 2022; Xu et al., 2022), and the beam-search variation of CQD (Arakelyan et al., 2021).

We refer to two methods with symbolic integration. We cannot reproduce the results since the codes for those two methods have not been released. However, since symbolic integration can also be applied to improve the LMPNN, we also list their results to show the potential.

**GNN-QE Zhu et al. (2022):** This requires 4 V100 GPU (32G), which is 8 times larger than the resources required by LMPNN. The official implementation has not been released.
**ENeSy (Xu et al., 2022):** The official implementation has not been released.

Table 9 shows that LMPNN is also compatible even with the symbolic integrated models at EPFO queries with only 1% trainable parameters at NELL and 10% trainable parameters at FB15k-237. For FB15k-237, there are still gaps between the neural CQA models and the models with symbolic

Table 8: Benchmark comparison with neural CQA models on NELL queries.

| Model | 1P | 2P | 3P | 2I | 3I | PI | IP | 2U | UP | 2IN | 3IN | INP | PIN | PNI | $A_P$ | $A_N$ |
|---|---|---|---|---|---|---|---|---|---|---|---|---|---|---|---|---|
| BETAE | 53.0 | 13.0 | 11.4 | 37.6 | 47.5 | 24.1 | 14.3 | 12.2 | 8.5 | 5.1 | 7.8 | 10.0 | 3.1 | 3.5 | 24.6 | 5.9 |
| ConE | 53.1 | 16.1 | 13.9 | 40.0 | 50.8 | 26.3 | 17.5 | 15.3 | 11.3 | 5.7 | 8.1 | 10.8 | 3.5 | 3.9 | 27.1 | 6.4 |
| MLP-Mix | 55.6 | 16.3 | 14.9 | 38.5 | 49.5 | 17.1 | 23.7 | 14.6 | 11.0 | 5.1 | 8.0 | 10.0 | 3.6 | 3.6 | 26.8 | 6.1 |
| Q2P | 56.5 | 15.2 | 12.5 | 35.8 | 48.7 | 22.6 | 16.1 | 11.1 | 10.4 | 5.1 | 7.4 | 10.2 | 3.3 | 3.4 | 25.4 | 5.9 |
| FuzzQE (ours) | 55.5 | 16.8 | 14.4 | 37.3 | 46.9 | 24.0 | 19.1 | 15.0 | 11.7 | 7.3 | 9.1 | 11.1 | 4.1 | 4.9 | 26.7 | 7.3 |
| FuzzQE (reported) | 58.1 | 19.3 | 15.7 | 39.8 | 50.3 | 28.1 | 21.8 | 17.3 | 13.7 | 8.3 | 10.2 | 11.5 | 4.6 | 5.4 | 29.3 | 8.0 |
| CQD | 60.8 | 18.3 | 13.2 | 36.5 | 43.0 | 30.0 | 22.5 | 17.6 | 13.7 | 0.1 | 0.1 | 4.0 | 0.0 | 5.2 | 28.4 | 1.9 |
| LMPNN | 60.6 | 22.1 | 17.5 | 40.1 | 50.3 | 28.4 | 24.9 | 17.2 | 15.7 | 8.5 | 10.8 | 12.2 | 3.9 | 4.8 | **30.7** | **8.0** |

Table 9: Comparison between LMPNN and symbolic integration methods. The number in brackets indicate the order of trainable parameters.

| Model | 1P | 2P | 3P | 2I | 3I | PI | IP | 2U | UP | 2IN | 3IN | INP | PIN | PNI | $A_P$ | $A_N$ |
|---|---|---|---|---|---|---|---|---|---|---|---|---|---|---|---|---|
| (FB15k-237) | | | | | | | | | | | | | | | | |
| LMPNN ($10^7$) | 45.9 | 13.1 | 10.3 | 34.8 | 48.9 | 22.7 | 17.6 | 13.5 | 10.3 | 8.7 | 12.9 | 7.7 | 4.6 | 5.2 | 24.1 | 7.8 |
| GNN-QE ($10^8$) | 42.8 | 14.7 | 11.8 | 38.3 | 54.1 | 31.1 | 18.9 | 16.2 | 13.4 | 10.0 | 16.8 | 9.3 | 7.2 | 7.8 | **26.8** | **10.2** |
| ENeSy ($10^8$) | 44.7 | 11.7 | 8.6 | 34.8 | 50.4 | 27.6 | 19.7 | 14.2 | 8.4 | 10.1 | 10.4 | 7.6 | 6.1 | 8.1 | 24.5 | 8.5 |
| (NELL) | | | | | | | | | | | | | | | | |
| LMPNN ($10^7$) | 60.6 | 22.1 | 17.5 | 40.1 | 50.3 | 28.4 | 24.9 | 17.2 | 15.7 | 8.5 | 10.8 | 12.2 | 3.9 | 4.8 | **30.7** | 8.0 |
| GNN-QE ($10^9$) | 53.3 | 18.9 | 14.9 | 42.4 | 52.5 | 30.8 | 18.9 | 15.9 | 12.6 | 9.9 | 14.6 | 11.4 | 6.3 | 6.3 | 28.9 | 9.7 |
| ENeSy ($10^9$) | 59.0 | 18.0 | 14.0 | 39.6 | 49.8 | 29.8 | 24.8 | 16.4 | 13.1 | 11.3 | 8.5 | 11.6 | 8.6 | 8.8 | 29.4 | **9.8** |

integrations. These results suggest that neural models can be potentially improved with symbolic integration. The additional cost is the larger computational cost.

We noticed that the task of answering logical queries are investigated over larger knowledge graphs (Ren et al., 2022). When considering larger knowledge graphs, neural CQA methods (discussed in the Appendix E) and symbolic integrated methods (discussed in this part) have different scalabilities. For neural CQA models, the intermediate embeddings are of fixed dimensions, while the sizes of intermediate fuzzy sets used in the symbolic integration methods grow linearly with the size of the knowledge graph. Such difference makes neural-symbolic methods more resource demanding and they may suffer from the scalabilities issues.

The differences between NELL and FB15k-237 can be explained by the quality of the ground knowledge graphs. However, integrating the symbolic method into neural CQA models and investigating the fundamental impact of ground KGs are beyond the scope of this paper. Our work connects the KG representation and neural CQA, which could also be combined with context and symbolic information. These extensions are left for future work and are expected to bring additional improvements.

