# OpenReview forum: "Logical Message Passing Networks with One-hop Inference on Atomic Formulas"
_ICLR.cc/2023/Conference — ICLR 2023 poster_

### Official Review · Reviewer_5THt · 2022-10-16

**Confidence:** 4
**Correctness:** 4
**Technical Novelty And Significance:** 3
**Empirical Novelty And Significance:** 2
**Recommendation:** 6

**Clarity, Quality, Novelty And Reproducibility:**

Clarity: The method is clearly written and easy to follow. But, I do find the authors did not include a discussion on how they handle the union operation using their message passing architecture. Also there are some weird indents at the start of many paragraphs, which look like uncleaned comments.

Quality & Novelty: I think the paper produces a nice combination of CQD and MPQE. The experiment seems lacking, the authors did not compare the proposed method with all recent baselines.

Reproducibility: the results seem reproducible with all the details in section 7.1.


**Strength And Weaknesses:**

The paper proposes a simple and effective method for query answering over knowledge graphs. It seems a combination of the ideas of CQD and MPQE. The authors perform extensive experiments and ablation studies to evaluate the proposed method. However, they did not seem to include the results of other recent baselines. Please find some more questions below.
- It misses some words in the first paragraph on page 4. “is able to produce a continuous truth value…”
- What does “suo” mean at the end of Sec 4.1?
- In Eq.5, what is the search domain for the embedding?
- How is the logical message passing neural network different from a regular message passing NN? The update function is almost exactly the same.
- The paper does not say how to handle the union operator. But I assume the way is to do a form of aggregation at the end. I suggest the authors add one paragraph about this.
- The paper mentioned one form of query that cannot be handled by the other methods? Does the proposed method handle such queries? How would you handle an atomic formula $\neg(a_1=a_2)$?
- How does the model perform on larger benchmarks in [1]?
- The baselines are not fully considered, there are GNNQE, kgTransformer and some other recently proposed query reasoning methods that are not compared in the main result.

[1] SMORE: Knowledge Graph Completion and Multi-hop Reasoning in Massive Knowledge Graphs

**Summary Of The Paper:**

The paper focuses on query answering over knowledge graphs. The idea is to use standard link prediction knowledge graph embeddings to model each triplet or atomic formula, create a message and spread it out to do a message passing over the query graph. Experiments on standard benchmark shows the proposed method achieves better or comparable results than prior state-of-the-art methods.

**Summary Of The Review:**

Please find the details above.

---

> ### Author Response · Authors · 2022-11-18
> **Feedback to Reviewer 5Tht**
>
> Dear Reviewer 5Tht,
>
> Thanks for your insightful comments. Here is our feedback.
>
> > In Eq.5, what is the search domain for the embedding?
>
> In Eq.5, the search domain is subjected to specific KG embeddings. It can be either a continuous (real or complex vector) space or a finite set of embeddings of entities. However, to obtain the closed-form message, we suggest using continuous spaces.
>
> > How is the logical message passing neural network different from a regular message passing NN? The update function is almost exactly the same.
>
> When devising LMPNN, the minimalism principle is chosen. Therefore, the ONLY difference between LMPNN and MPNN is the ``logical’’ message to be passed. In this way, this work reveals a piece of simple but effective information that passing the logical message is essential to perform logical reasoning on knowledge graphs.
>
> In addition, we invite the reviewers to read the updated Appendix D, where we updated two general constructions of the closed-form logical messages for various knowledge graph embeddings and how six pretrained KG embeddings (ComplEx, RotatE, ConvE, RESCAL, TransE, and DistMult) performs on complex query answering tasks.
>
> Therefore, LMPNN can be applied to leverage many KG representations by using logical messages.
>
> We also expect that more advanced graph neural network designs will bring stronger predictive power. But such a design is beyond the scope of this paper.
>
> > The paper does not say how to handle the union operator. But I assume the way is to do a form of aggregation at the end. I suggest the authors add one paragraph about this.
>
> We have mentioned how to solve DNF queries (which contain logical disjunctions). In the paragraph before Section 4.1, we stated *“Solving A[Q, KG] is equivalent to solving the answer sets for all conjunctive queries”*. Moreover, In the first paragraph of Section 6.3 “Answering complex queries with LMPNN”, we stated that *“Firstly, we estimate the free variable embedding for each conjunctive query. Then, we retrieve the answer entities by multiple free variable embeddings as in the previous works”*.
>
> In the updated version, we elaborate more on handling the logical disjunctions.
>
> > The paper mentioned one form of query that cannot be handled by the other methods? Does the proposed method handle such queries? How would you handle an atomic formula ¬(a1=a2) ?
>
> Thanks for mentioning the query types. We discuss how to handle atomic formulas with logical equality in Appendix A. Logical equality is a predicate, and the key is to determine $f(a, equality)$, which is essential as indicated in Appendix D.1. Specifically, we can choose $f(a, equality) = a$ since two equal entities should have the same embedding. Then the logical message passed through negated atomic formula $\lnot (a_1 = a_2)$ can be derived by Proposition 2 in Appendix D.1.
>
>
> > How does the model perform on larger benchmarks in [1]?
>
> Thanks for mentioning this work. Given that this work does not release its dataset, we cannot make quantitative comparisons. However, we could make the following conclusion about the LMPNN framework given the current information:
>
> LMPNN is scalable. LMPNN is built upon a given pretrained KG embedding. Therefore, the computational cost of the forward pass of LMPNN only depends on the KG embedding and the size of the query graph, which is independent of the sizes of knowledge graphs. Compared to GNN-QE which uses fuzzy sets as the intermediate states in GNN, the size of its space cost is O(|E|) (E indicates the set of entities), which grows linearly with the sizes of knowledge graphs.
> LMPNN’s performance is affected by the backbone knowledge graph embeddings (please refer to our evaluation of six backbone KGEs). Therefore, we could expect that more advanced KG embedding for the large knowledge graphs could bring better performances of LMPNN.
>
> > The baselines are not fully considered, there are GNNQE, kgTransformer and some other recently proposed query reasoning methods that are not compared in the main result.
>
> We have compared ENeSy (NeurIPS 2022), Q2P (NAACL 2022), GNN-QE (ICML 2022), MLP-Mix (ICLR 2022), FuzzQE (AAAI 2022), ConE (NeurIPS 2021), CQD (ICLR 2021), and BetaE (NeurIPS 2020) in the Appendix. GNN-QE has also already been considered and compared in Appendix F.
>
> kgTransformer (KDD 2022), as mentioned by the reviewer, cannot handle negation queries, and is not evaluated on the existing datasets. So we already have a direct advantage over kgTransformer (for handling negation queries), and we don’t have existing scores to refer to. We discuss this work in Appendix E.
>
> Best regards,
>
> Authors of Paper4815

---

### Official Review · Reviewer_zhhh · 2022-10-21

**Confidence:** 5
**Correctness:** 3
**Technical Novelty And Significance:** 4
**Empirical Novelty And Significance:** 3
**Recommendation:** 6

**Clarity, Quality, Novelty And Reproducibility:**

## Clarity
This paper is clearly written and well organized

## Quality
This paper is technically sound. The main claims are well supported by theoretical analyses or experiments, and the experimental results are convincing.


## Novelty
The overall framework seems novel and interesting to me. The idea of exploiting the power KGE to perform logical reasoning is inspired by CQD (ICLR21), while the authors further incorporate message passing to the framework and succeed to model logical negation.

## Reproducibility
The authors provide detailed experimental settings and the range of grid search, so the reproducibility looks nice.

**Strength And Weaknesses:**

## Strength
1. This paper manages to answer complex EFO-1 queries by exploiting the potential of pretrained knowledge graph embeddings, showing that pretrained embeddings also contain abundant semantic information for logical negation.
2. The proposed LMPNN is simple but effective, attaining new SOTA results on benchmark datasets.
3. The authors conduct extensive ablation studies to evaluate the influence of different hyper-parameters.
4. This paper is clearly written and easy to follow.

## Weakness
1. The proposed framework is dependent on the underlying knowledge graph embeddings. The authors may want to provide more results using other KGE models (e.g., DistMult, TransE, RotatE, etc).
2. To perform efficient logical message passing, it seems that a closed-form message encoding function is necessary. If the underlying KGE is TransE or ConvE, then how to implement PMPNN?
3. The authors define a class of more expressive logical queries in Definition 1 and show that it can represent more queries compared with operator trees. However, there is no such kind of query in the current benchmark datasets.


**Summary Of The Paper:**

The authors propose to represent first-order logical queries as query graphs and then design a novel message passing-based framework LMPNN on the graphs to answer given queries. This is a successful attempt to combine the power of pretrained knowledge graph embeddings and message passing to perform logical reasoning. The proposed LMPNN is quite simple (a virtue) but experiments demonstrate its superior performance.

**Summary Of The Review:**

This paper proposes a novel and interesting model for logical reasoning. The proposed model is simple and shows strong performance on standard benchmarks. The submission will be more solid if analyses and experiments for more KGE models can be provided.

---

> ### Author Response · Authors · 2022-11-18
> **Feedback to Reviewer zhhh**
>
> Dear reviewer zhhh,
>
> Thanks for your informative suggestions. Here is our feedback for the backbone KGE models.
>
> > The authors may want to provide more results using other KGE models (e.g., DistMult, TransE, RotatE, etc.).
>
> In the updated version, we also provide closed-form logical message encoding functions and empirical results of DistMult, TransE, RotatE, ConvE, RESCAL, and ComplEx backbones in Appendix D.2. We use the pretrained KG embeddings checkpoints released by (Ruffielli et al., 2020).
>
> The empirical results show that LMPNN could provide decent results on various queries with various KG representations of low dimensions. It is much better than CQD on negation queries and compatible with recent neural CQA methods. It is suggested that LMPNN is generally a good way to leverage the predictive power of knowledge graph embedding, even though the performances on LMPNN + [specific kind of KG embedding] are related by the backbone KG embeddings.
>
>
> > To perform efficient logical message passing, it seems that a closed-form message encoding function is necessary. If the underlying KGE is TransE or ConvE, then how to implement PMPNN?
>
> In the updated version, we discuss two general ways to construct the closed-form message encoding function for two types of KG embeddings, i.e., KGE with inner-product scoring functions (ComplEx, DistMult, ConvE, and RESCAL) and KGE with distance scoring function (RotatE and TransE) in the Appendix D.1. These two scoring functions cover a large portion of existing KG embeddings. We also demonstrate how to derive closed-form message encoding functions by applying these two constructions to the six aforementioned KGEs (see Appendix D.2). The constructed function is simple but empirically effective.
>
> > The authors define a class of more expressive logical queries in Definition 1 and show that it can represent more queries than operator trees. However, there is no such kind of query in the current benchmark datasets.
>
> The construction of a new benchmark is beyond the scope of this paper. However, in the updated Appendix A, we discuss how to use LMPNN to solve such queries. We believe that the LMPNN is at least an actionable and the first solution to answer logical queries in such a general form.
>
>
> Best regards,
>
> Authors of Paper4815
>
>
> ## Reference:
>
> Ruffinelli, D., Broscheit, S., & Gemulla, R. (2020). You can teach an old dog new tricks! on training knowledge graph embeddings.

---

### Official Review · Reviewer_NhRN · 2022-10-24

**Confidence:** 3
**Correctness:** 3
**Technical Novelty And Significance:** 3
**Empirical Novelty And Significance:** 3
**Recommendation:** 6

**Clarity, Quality, Novelty And Reproducibility:**

The paper is written clearly and technically sound. The proposed method (w.r.t one-hot inference-based GNN) is novel.

**Strength And Weaknesses:**

Strength:

1. LMPNN is quite parameter-efficient, where only MLP and the embeddings of existential variable and free variable are learned.

2. Due to the novel design of closed-form approximation of single-hop ComplEx, LMPNN is inference-efficient (in a GNN manner) compared to optimization-based methods like CQD [1].

3. LMPNN achieves competitive or better performance on FB15k, FB15k-237, and NELL datasets.

4. Compared to previous work CQD, LMPNN can additionally model the negation operation.


Weaknesses:

1. My main concern is on the comparison with the optimization-based method (i.e., CQD). The author claims that "Moreover, it is unclear whether CQD can be applied to complex queries with negation operators". Could I interpret it as the CQD is evaluated without the negation operations for INP, PIN, PNI, etc settings? Would it be possible to just use Eq. 7&8 in the paper to get an optimization-based baseline? Since one of the contributions in the paper is a GNN-like inference, I believe a fair comparison with optimization-based methods is necessary.

2. The method is motivated by how to better perform one-hop queries, however, the experimental results show that LMPNN is usually outperformed by CQD in the 1P setting. Since in 1P, the one-hop ComplEx closed-form approximation should lead to the same solution as CQD, could the author explain the reasons?

3. I feel the claim of bridging the gap between EFO-1 query answering and "the long-standing achievements of KGR" is a little over-claimed, given that previous work like CQD already uses the exactly same pre-trained KGE model (ComplEx).

Typos:

Page 5, " In Figure 1, the central node x receives messages from all neighbor nodes, suo." what does "suo" mean?

**Summary Of The Paper:**

The paper proposes a Logical Message Passing Neural Network (LMPNN), which relies on pre-trained knowledge graph embeddings and MLP-based local one-hop inference to perform the Complex Query Answering (i.e., EFO-1) task. Compared to the prior work CQD [1], which formalizes the KG reasoning as an optimization problem, this paper proposes to use the closed-form solution of ComplEX to approximate the single-hop inference and use an MLP-based multi-layer GNN to perform the multi-hop reasoning.

[1]: Arakelyan, Erik, Daniel Daza, Pasquale Minervini, and Michael Cochez. "Complex Query Answering with Neural Link Predictors." In International Conference on Learning Representations. 2020.

**Summary Of The Review:**

The proposed LMPNN is 1) parameter-efficient, 2) inference-efficient (compared to optimization-based methods like CQD), and 3) achieves competitive or better performance on three EFO-1 datasets. My main concern is about the fair comparison with optimization-based methods with negations. Therefore, I can only recommend a weak accept for the paper.

---

> ### Author Response · Authors · 2022-11-18
> **Feedback to Reviewer NhRN**
>
> Dear reviewer NhRN,
>
> Thank you for your insightful suggestions. Here is our feedback regarding the difference between CQD and LMPNN.
>
> > My main concern is on the comparison with the optimization-based method.
>
> In the original submission, the reported CQD scores on negation queries are derived by introducing the fuzzy logic negator n(x) = 1 - x (shown in Eqs. 7&8) to the optimization objective. Therefore, this baseline is fairly compared.
>
> In the updated version, the extended CQD with fuzzy logical negator is renamed as CQD(E) to be different from the CQD. Details of CQD(E) can also be found in Appendix B. Moreover, a possible reason why CQD(E) performs poorly on negation queries is the non-convexity of the optimization objective. We reveal the non-convexity in Appendix B.1.
>
> > However, the experimental results show that LMPNN is usually outperformed by CQD in the 1P setting.
>
> We train an MLP layer in LMPNN for all query types. Thus, the MLP is not particularly optimized to overfit 1P queries. Cost-sensitive learning can be applied here to improve 1P results. However, it may overfit 1P samples.
>
> > I feel the claim of bridging the gap between EFO-1 query answering and "the long-standing achievements of KGR" is a little over-claimed, given that previous work like CQD already uses the exactly same pre-trained KGE model (ComplEx).
>
> Thanks for raising this concern. Here we provide some more discussion. First, LMPNN solves EFO-1 query answering tasks with strong performances while CQD(E) performs poorly on negation queries even extended with fuzzy logic negator. Thus, the existing optimization-based framework cannot apply KG representations to solve the *entire* EFO-1 query-answering task. One possible reason for the failure of CQD is discussed in Appendix B.1.
>
> Second, as we introduced in the general feedback, we show that LMPNN can also adapt to various KG representations. In addition to ComplEx and DistMult discussed in the CQD paper, we include RotatE, TransE, ConvE, and RESCAL in Appendix D.2. Our theoretical construction in D.1 shows the potential to include more KG representations
>
> Therefore, we believe we provide strong empirical clues for connecting KG representations to the *entire* EFO-1 query answering with LMPNN.
>
>
> Best regards,
>
> Authors of Paper4815

---

### Author Response · Authors · 2022-11-18
**General Feedback**

Dear Reviewers,

The authors thank three reviewers for their constructive suggestions. Three reviewers acknowledged the novelty of this paper but also raised three major concerns. Those concerns are all addressed in the updated version. The modified parts are highlighted.

Reviewer NhRN mentioned that
> My main concern is about the fair comparison with optimization-based methods with negations.

We compared the extended CQD version with the fuzzy logic negator in the original version. So the comparison is fair in terms of fuzzy logic negator. We invite reviewers to read Appendix B, where we showed how to extend the optimization-based methods (CQD) to the negation queries by incorporating the fuzzy logic negation into the objective function.

Reviewer zhhh suggested that
> The submission will be more solid if analyses and experiments for more KGE models can be provided.

We invite reviewers to read the new Appendix D.
- In Appendix D.1, we summarized the general closed-form logical message for KG embeddings trained by the inner-product-based scoring function and those by the distance-based scoring function. In this way, we presented clear theoretical guidelines and executable practices for using KG embeddings in LMPNN.
- In Appendix D.2, we apply our analysis to all knowledge graph embeddings mentioned in reviewer zhhh’s comment, including ConvE, DistMult, RotatE, TransE, and RESCAL. Combined with ComplEx, six kinds of closed-form logical messages passed in LMPNN are all presented.
- In Appendix D.3, we also presented experimental results on ConvE, DistMult, RotatE, RESCAL, and TransE embeddings on the FB15k-237 dataset. The pretrained KG embeddings are downloaded from https://github.com/uma-pi1/kge. The empirical results show that stronger KG representations produce better complex query-answering performances.

Reviewer 5Tht commented that
> The experiment seems lacking, the authors did not compare the proposed method with all recent baselines.

We invite the reviewers to read Appendix E and Appendix F. As far as we have concerned, we have already considered a long list of published baselines before the paper submission, including
- ENeSy (NeurIPS 2022),
- Q2P (NAACL 2022),
- GNN-QE (ICML 2022),
- MLP-Mix (ICLR 2022),
- FuzzQE (AAAI 2022),
- ConE (NeurIPS 2021),
- CQD (ICLR 2021),
- BetaE (NeurIPS 2020).

Specifically, GNN-QE mentioned by reviewer 5Tht has already been presented in Appendix F in the original submission. Meanwhile, kgTransformer (KDD 2022), also mentioned by reviewer 5Tht, cannot handle negation queries. So kgTransformer is not compared. In this way, we think the baselines included are comprehensive and sufficient.  For the larger benchmark SMORE (KDD 2022), the dataset is not released, so we are not able to run that directly.

However, the empirical evaluation in Appendix D.3 shows stronger backbone KG representation leads to stronger LMPNN performances. So one could expect that the problem of improving the performance of LMPNN, even on larger graphs, partially boils down to the problem of improving the backbone KG representations. More discussion about the scalability of LMPNN is also stated in Appendix F and the direct feedback to Reviewer 5Tht.

By answering three constructive suggestions, LMPNN is shown to be an effective way (by comparing to a comprehensive list of baseline methods) to exploit various forms of KG embeddings (two types of KG representations characterized by two general types of scoring functions in Appendix D.1 and 6 exemplar KG representations in Appendix D.2) to answer complex queries, especially those with logical negation (much stronger than the extended CQD).

To summarize, this paper conveys several simple but very powerful takeaways

- One-hop inference is powerful: Pretrained  KG embeddings contain essential one-hop information. Such information is formulated as the solutions of four one-hop inference problems and is critical for knowledge graph reasoning.
- Query graph representation is more expressive: Operator tree formulation for complex (multi-hop) queries only represents an incomplete subset of first-order logical queries. Query graphs represent a more general set of queries, and LMPNN can solve them.
- LMPNN bridges the **one-hop** information and **multi-hop** complex query answering tasks and reaches state-of-the-art performance by passing the “correct” messages, which are the approximate closed-form solutions of one-hop inference.

Best regards,

Authors of Paper4815

---

> ### Author Response · Authors · 2022-12-10
> **More comment on MPQE**
>
> Dear reviewers,
>
> We find an interesting line of research that also uses message-passing networks to answer queries on knowledge graphs, i.e. Message Passing Query Embedding (MPQE) (Daza and Cochez, 2020). This abbreviation is mentioned by reviewer 5THt. We are sorry about figuring out what it refers to recently.
>
> Here is our justification for differences and similar features between MPQE and LMPNN.
>
> The key differences are
> - What queries can be answered?
>   - MPQE can only answer EPFO queries.
>   - LMPNN can answer first-order queries, which include EPFO queries as a **strict subset**.
>   - This difference is caused by the different designs of the message-passing mechanisms discussed below.
> - What message is passed?
>    - MPQE employs R-GCN, where the message is computed by the neighboring entities and the relations in between. The parameters of the message encoding function **MUST** be learned during the training process.
>   - LMPNN passes the message computed by closed-form results of one-hop inferences, which include the information from neighboring entities, connecting relations, and logical negations. **NO** parameter of the message encoding function will be learned. LMPNN can accept a broad range of knowledge graph embeddings, as long as they accept distance or inner product as its scoring function. This makes LMPNN very flexible.
>
> Similar features are
> - Both MPQE and LMPNN use the cosine similarity to retrieve the answer embeddings.
> - Both MPQE and LMPNN define the dynamic depths of GNN layers by the statistics from the query graph. In MPQE, it is the **diameter** of the query graph. In LMPNN, it is the largest distance between the free variable and the constant entities.
>
> To summarize, MPQE shares properties similar to LMPNN, such as cosine similarities and dynamic depth of GNN layers. However, LMPNN is more powerful, efficient, and flexible to leverage existing studies in knowledge graph representations, given the difference discussed above.
>
> MPQE will be acknowledged as an important predecessor of LMPNN in the next version of the paper.
>
> Best regards,
>
> Authors of Paper4815
>
> ## Reference
>
> Daza, D., & Cochez, M. (2020). Message passing query embedding. ICML 2020 Workshop on Graph Representation Learning and Beyond. arXiv preprint arXiv:2002.02406.

---

### Decision · Program_Chairs · 2023-01-20

**Decision:**

Accept: poster

**Justification For Why Not Higher Score:**

I believe the reviewers could've increased the scores slightly after reviewing the authors' updated version.

**Justification For Why Not Lower Score:**

The paper is solid and the approach is novel.

**Metareview: Summary, Strengths And Weaknesses:**

Summary: The paper addresses the problem of answering complex queries (EFO-1 - Existentially quantified First Order queries and has a single free variable) over a knowledge graph. The proposed approaches, LMPNN, uses standard link prediction embeddings to model atomic formulae and does message passing over the query graph. Experiments on standard benchmark datasets showed that LMPNN either performed better or comparably to SoTA methods.

Strengths: All reviewers acknowledged that the paper is well-written and easy to follow. The proposed approach is novel, simple and yet effective in practice. While reviewers raised several minor concerns and clarification questions in their initial comments, I believe that the authors have done a good job addressing them, both in the rebuttal and in the updated version of the paper (with extended and detailed appendix sections).

Weaknesses: The problem addressed in this paper, while very interesting, might be interesting to a small community. While the method shows solid results on standard benchmark datasets, its effectiveness in real-world applications remains to be seen.

**Note From Pc:**

if the above contains the word "oral" or "spotlight" please see: "oral" presentation means -> notable-top-5% and "spotlight" means -> notable-top-25%. As stated in our emails, we are disassociating presentation type from AC recommendations